# X-linked myopathy with excessive autophagy: characterization and therapy testing in a zebrafish model

Lily Huang [1,2,7], Rebecca Simonian[1,7], Michael A Lopez [3,4], Muthukumar Karuppasamy [3,4], Veronica M Sanders[3], Katherine G English[3], Lacramioara Fabian [1], Matthew S Alexander [3,4]✉ & James J Dowling [1,2,5,6]✉

## Abstract

X-linked myopathy with excessive autophagy (XMEA), a rare childhood-onset autophagic vacuolar myopathy caused by mutations in *VMA21*, is characterized by proximal muscle weakness and progressive vacuolation. *VMA21* encodes a protein chaperone of the vacuolar hydrogen ion ATPase, the loss of which leads to lysosomal neutralization and impaired function. At present, there is an incomplete understanding of XMEA, its mechanisms, consequences on other systems, and therapeutic strategies. A significant barrier to advancing knowledge and treatments is the lack of XMEA animal models. Therefore, we used CRISPR-Cas9 editing to engineer a loss-of-function mutation in zebrafish *vma21*. The *vma21* mutant zebrafish phenocopy the human disease with impaired motor function and survival, liver dysfunction, and dysregulated autophagy indicated by lysosomal de-acidification, the presence of characteristic autophagic vacuoles in muscle fibers, altered autophagic flux, and reduced lysosomal marker staining. As proof-of-concept, we found that two drugs, edaravone and LY294002, improve swim behavior and survival. In total, we generated and characterized a novel preclinical zebrafish XMEA model and demonstrated its suitability for studying disease pathomechanisms and identifying potential therapeutic targets.

**Keywords** Myopathy; XMEA; VMA21; Autophagy; Zebrafish
**Subject Categories** Genetics, Gene Therapy & Genetic Disease; Methods & Resources; Musculoskeletal System

## Introduction

Autophagy is the cellular recycling system of the body mediated by lysosomes, crucial for maintaining homeostasis by degrading unnecessary and/or dysfunctional cellular components. The vacuolar hydrogen ion ATPase (V-ATPase) complex is responsible for lysosomal acidification required to activate proteases for protein and organellar breakdown (Forgac, 2007). Consequently, impaired lysosomal acidification is responsible for a wide range of multi-system human disorders linked to disrupted autophagy (Cocchiaro et al, 2023).

The X-linked vacuolar ATPase Assembly Factor 21 (*VMA21*) gene encodes the vacuolar ATPase assembly integral membrane protein, essential for the assembly of the V-ATPase and its translocation to the lysosome (Hill and Stevens, 1994). Mutations in this gene are rare, manifesting in affected male individuals. The first phenotype described related to variations in *VMA21* was X-linked myopathy with excessive autophagy (XMEA, OMIM 310440, *Xq28*) (Dowling et al, 2015; Ramachandran et al, 2013), a rare recessive disease with childhood onset. Interestingly, and in contrast to other disorders that impair lysosomal acidification resulting in multiple affected organ systems, XMEA predominantly presents as a mild-moderate skeletal muscle disease characterized by progressive proximal muscle weakness and atrophy, as well as pathology that includes the presence of double-membrane vacuoles containing extracellular matrix and membrane components (Ramachandran et al, 2013). The consequences, if any, of variations in *VMA21* on other systems are not well known, but there have been rare observations of heart involvement and liver dysfunction, with one report of congenital disorder of glycosylation with autophagic liver disease (Yan et al, 2005; Ruggieri et al, 2015; Blanco-Arias et al, 2023; Munteanu et al, 2017). While all myopathy-associated mutations described in *VMA21*, a gene with only 3 exons, are associated with reduced or absent protein expression (Ramachandran et al, 2013; Munteanu et al, 2015;

[1]Program for Genetics and Genome Biology, Hospital for Sick Children, Toronto, ON M5G 1E8, Canada. [2]Department of Molecular Genetics, University of Toronto, Toronto, ON M5S 3K3, Canada. [3]Department of Pediatrics, Division of Neurology at the University of Alabama at Birmingham and Children's of Alabama, Birmingham, AL 35294, USA. [4]Department of Genetics at the University of Alabama at Birmingham, Birmingham, AL 35294, USA. [5]Division of Neurology, Hospital for Sick Children, Toronto, ON M5G 1E8, Canada. [6]Department of Paediatrics, University of Toronto, Toronto, ON M5G 1E8, Canada. [7]These authors contributed equally as first authors: Lily Huang, Rebecca Simonian. ✉E-mail: matthewalexander@uabmc.edu; james.dowling@sickkids.ca

Crockett et al, 2014; Ruggieri et al, 2015), there is likely a genotype-phenotype continuum related to the extent to which mutations alter protein expression and/or function and disease severity. In general, XMEA is considered a mild, slowly progressive condition, though there is important clinical variability (Fernández-Eulate et al, 2024), including a subset of patients with extremely severe disease (including death in early childhood) (Blanco-Arias et al, 2023).

While V-ATPases are ubiquitous with crucial roles in all cell types, in XMEA skeletal muscle is the main clinically affected organ (Munteanu et al, 2015). At present, there is limited knowledge on mechanisms underlying how *VMA21* mutations, and the subsequent defects in lysosome function and autophagy, result in skeletal muscle disease. In addition, and stemming from this gap in knowledge, there are currently no therapeutic strategies for this disease. A significant barrier in the field is the lack of preclinical animal models that recapitulate the genetics (i.e., loss of *VMA21* expression) and the phenotypic changes observed in patients, encouraging the development of novel animal models. In recent years, the zebrafish (*Danio rerio*) is recognized as a powerful preclinical model in general and specifically with regard to muscle and autophagy research. In general, zebrafish are advantageous owing to their low husbandry costs, rapid *ex utero* development, large clutch sizes (~100–300 embryos per cross), optical clarity during development, and ease of genetic manipulation and use for drug testing (Karuppasamy et al, 2024; Gibbs et al, 2013). More specific to the study of XMEA and other autophagy-related muscle disorders, zebrafish have high genetic similarity to humans with key mammalian autophagy-related genes being present in their genome, suggesting conservation of autophagy pathways (Moss et al, 2020). In addition, they share similar muscle physiology to humans and are able to, in some cases, better recapitulate the severity of human skeletal muscle disorders than corresponding mouse models (Gibbs et al, 2013; Moss et al, 2020).

In this study, we aimed to generate and characterize the first zebrafish model with loss of expression mutations in *vma21*. Our mutant model accurately presents multiple features of the severe form of XMEA, including progressive impairment of movement, early lethality, liver dysfunction, and cellular pathologic features including neutralization of the lysosome and the presence of characteristic vacuoles in skeletal muscles. As proof-of-concept, we tested several chemicals that modulate autophagy, and found two that effectively improve swim behavior and prolong survival. In total, we have established a novel zebrafish model of *vma21* deficiency and demonstrated its utility for studying disease pathogenesis and identifying therapies.

# Results

## Establishing a zebrafish model of XMEA

There is a single ortholog of *vma21* in zebrafish which shares 70% identify at the protein level with the human gene. Using CRISPR-Cas9 gene editing, and a guide RNA (gRNA) targeted to exon 2 of the zebrafish *vma21* locus, two loss-of-function mutations in the zebrafish *vma21* gene were generated (Fig. 1A,B). The first, referred to as *vma21*$^{Δ1}$, harbors a 1 base pair (bp) deletion that results in a frameshift mutation without a premature stop codon. The second, referred to as *vma21*$^{Δ14ins21}$, has a 14 bp deletion and a 21 bp

insertion. A new stop codon is introduced within the 21 bp insertion. Zebrafish harboring mutations on both alleles of the *vma21* gene are herein referred to as *vma21* mutants. Western blot analysis confirmed Vma21 protein levels to be decreased in both mutant models as compared to wild-type (WT) and respective heterozygous (HET) control larvae at 5 days post fertilization (dpf) (Fig. 1C,D).

## vma21 mutants have overt phenotypic differences compared to controls including abnormal morphology, impaired swim behavior, and reduced survival

We characterized the resulting phenotype of biallelic *vma21* mutant zebrafish. *vma21*$^{Δ1/Δ1}$, *vma21*$^{Δ14ins21/Δ14ins21}$, and *vma21*$^{Δ1/Δ14ins21}$ fish were all initially examined, and all were found to produce identical phenotypes (Fig. 2A–D). Thus, subsequent downstream investigations were mostly performed on *vma21*$^{Δ1/Δ1}$ mutants. *vma21* mutants display overt morphological differences compared to WT and HET sibling controls including reduced pigmentation, reduced body length, and non-inflated swim bladders (Fig. 2E,F). These findings are consistent with zebrafish mutants that have altered skeletal muscle structure and function.

We next investigated survival of *vma21* mutant zebrafish (Fig. 2G). As compared to WTs, which can survive 2+ years, death was first observed in *vma21* mutants at 8 days post fertilization (dpf). The majority of mutant fish die between 9 and 11 dpf, with none surviving beyond 12 dpf. This indicates a severe reduction in viability associated with almost complete loss of *vma21*. While premature death is not often seen in humans with mutations in *VMA21*, it has been noted that no patients have complete loss-of-function mutations in this gene, and if they did, it would more than likely be lethal (Ruggieri et al, 2015). Moreover, severe cases have been reported, including cases of neonatal death (Pegat et al, 2022) and of severe weakness with death in the first year of life (Blanco-Arias et al, 2023).

Lastly, we measured swim behavior as a surrogate for muscle function. Starting at 5 dpf, *vma21* mutants had obvious reductions in movement exemplified by decreased ability to swim away from a stimulus, with 100% of mutants being considered "low responders" in the touch-evoked escape response assay. To further quantify movement, we utilized the ZebraBox platform, an automated open field tracking system (Volpatti et al, 2022), under both free-swimming conditions (Fig. 2H–J) and upon stimulation with optovin (Fig. 2K–M), a photoactive compound that stimulates motor behavior in zebrafish larvae (Kokel et al, 2013). With both assays, we detected significant reductions in time spent moving and total distance traveled for both *vma21* mutants. The *vma21* mutants also showed a significant reduction in average swim velocity (except for *vma21*$^{Δ1/Δ1}$) in the optovin swim assay. Overall, these findings are consistent with muscle-derived impairment in motor activity, the zebrafish equivalent of reduced ambulation in humans (Smith et al, 2013).

## vma21 mutants have abnormal lysosomal acidification and changes consistent with aberrant autophagy

The anticipated cellular consequence of loss of *VMA21* expression is failure of the lysosomes to acidify. To test for this, we used the LysoTracker Red reagent, which selectively marks acidic organelles,

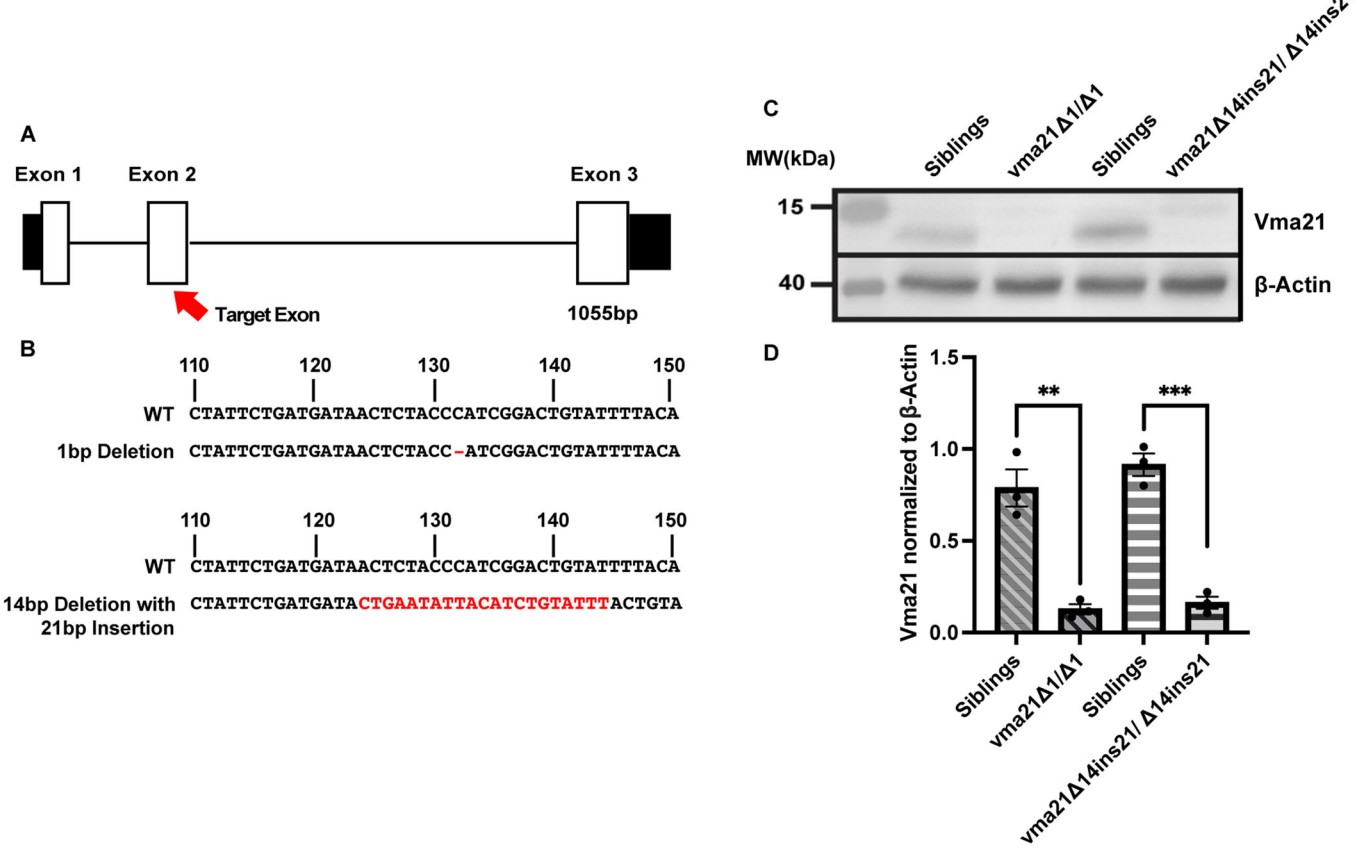

**Figure 1. Generation and validation of *vma21* mutant zebrafish.**

(A) Schematic of the zebrafish (*Danio rerio*) *vma21* gene. Mutations in exon 2 of *vma21* were created using CRISPR-Cas9 gene editing. (B) Two mutant lines were generated: (1) a 1 base pair (bp) deletion and (2) a 14 bp deletion with a 21 bp insertion. A compound heterozygous line was also generated by crossing the two mutant lines together. (C) Representative western blot of Vma21 and b-actin (housekeeping protein) protein levels in both the *vma21* 1 bp deletion (vma21^Δ1/Δ1) and the 14 bp deletion with 21 bp insertion (vma21^Δ14ins21/Δ14ins21) homozygous mutants compared to their respective wild-type/heterozygous sibling controls at 5 days post fertilization. Vma21 is ~11 kDa in size and b-actin is ~42 kDa. (D) Quantitative analysis of Vma21 western blot normalized to b-actin loading controls shows that Vma21 protein levels are significantly reduced in both *vma21* homozygous mutants (fivefold less) compared to their respective sibling controls. Each western blot had a total of three biological replicates. For each sample, $n = 30$ zebrafish were utilized. All values are shown as mean ± SEM. Significance at **$P < 0.01$; ***$P < 0.001$ (unpaired two-tailed $t$ test). Source data are available online for this figure.

on whole-mount 5 dpf zebrafish larvae (Fig. 3A–C). In WT fish, robust punctate staining within the muscle compartment, consistent with enumeration of acidic lysosomes, was observed. In contrast, in *vma21* knockouts, a complete absence of LysoTracker Red staining was observed, indicating a failure of organelles to acidify in the absence of *vma21*. We next performed double immunofluorescence staining in myofibers for lysosome-associated membrane protein-1 (Lamp1), a lysosomal marker, and dystrophin, expressed in differentiated myofibers, of *vma21* mutants and WTs at 5 dpf (Fig. 3D–F). Quantification of fluorescence intensity at the myotendinous junctions (MTJs, Fig. 3G) and in the entire myofiber (Fig. 3H) shows significantly reduced levels of Lamp1 in *vma21* mutants, suggestive of reduced lysosomal biogenesis within the myofiber. Taken together, these findings highlight impaired lysosomal activity and function with a loss of *vma21*.

A downstream consequence of lysosomal dysfunction is the presence of autophagic vacuoles in patient skeletal muscle, a classic hallmark of XMEA, which are best appreciated on ultrastructural analysis. We performed electron microscopy on WT and *vma21*

mutants at 6 dpf. No vacuoles were observed in WT myofibers; in contrast, easily identifiable, though infrequent, vacuoles with electron-dense material and naked membranes within the vacuole walls were seen in *vma21* mutants, properties consistent with autophagic vacuoles (Fig. 4A,B).

The presence of autophagic vacuoles is suggestive of aberrant autophagy. Therefore, we next measured the levels of LC3, an autophagy marker, by western blot from whole fish protein extracts. As a proof-of-concept positive control, we first performed a dose-escalation ethanol (EtOH) treatment in WT zebrafish larvae which has previously been shown to induce whole-body autophagy (Wen et al, 2022; Varga et al, 2015). As expected, we observed greater LC3I protein expression, and thus reduced LC3II/LC3I ratio, with increases in ethanol toxicity past the 0.5% limit, suggestive of increased autophagy (Fig. EV1). Similarly, in *vma21* mutants as compared to WT clutchmates, we observed a significant increase in LC3I and LC3II expression and a corresponding decrease in the LC3II/LC3I ratio, consistent with disruption of autophagic flux (Fig. 4C,D). As further proof of impaired autophagy with mutations

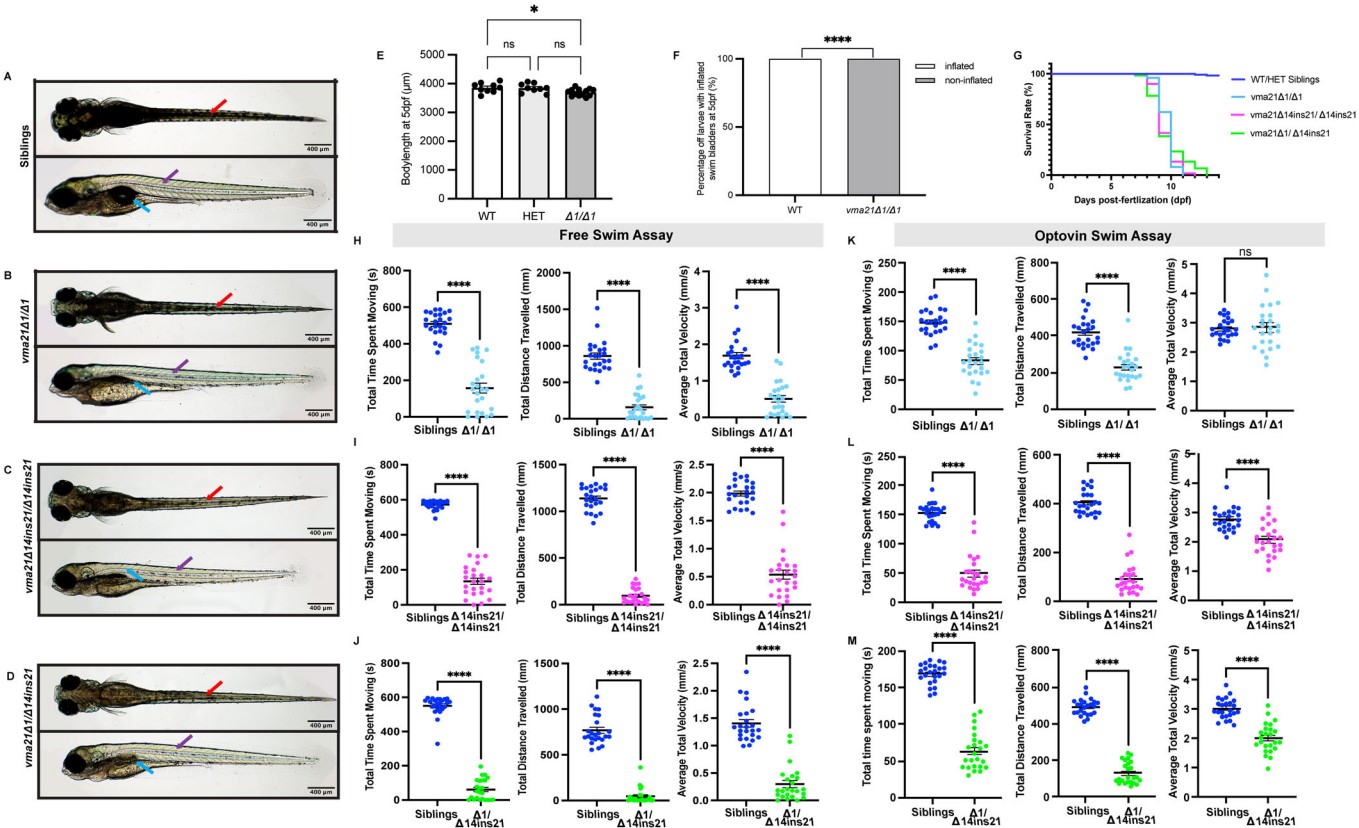

**Figure 2. *vma21* mutant zebrafish display overt morphological differences compared to controls.**

Representative light microscopy images of (**A**) wild-type (WT)/heterozygous (HET) siblings (**B**) *vma21^Δ1/Δ1^* (homozygous for a 1 base pair (bp) deletion in *vma21*), (**C**) *vma21^Δ14ins21/Δ14ins21^* (homozygous for a 14 bp deletion with a 21 bp insertion in *vma21*), and (**D**) *vma21^Δ1/Δ14ins21^* (a compound heterozygous line that contains both aforementioned mutations in *vma21*) zebrafish mutants in both the ventral and lateral positions at 5 days post fertilization (dpf). The red arrows on the panel showing the larvae in the ventral position indicate the greater level of melanophores (darker pigmentation) in the siblings in comparison to all the mutant lines. There is a decrease of xanthophores (the yellowish pigment) in the mutant lines (purple arrows) in comparison to all the siblings. The blue arrows show reduced swim bladder size within the mutant lines. Scale bars: 400 mm. (**E**) Bar graph showing a significant difference in body length between WT and *vma21^Δ1/Δ1^* mutants but not between WT and HETs or HETs and *vma21^Δ1/Δ1^* mutants at 5 dpf. n = 9–13/group. Data represented as mean ± SEM. Significance at *P < 0.05 (one-way ANOVA with Tukey's multiple comparisons test). (**F**) Stacked bar graph of the percentage of WT vs *vma21^Δ1/Δ1^* mutants with inflated swim bladders at 5 dpf. n = 10–13/group. Significance at ****P < 0.0001 (Fisher's exact test). (**G**) Kaplan–Meier curve showing percent survival of WT and *vma21* mutant zebrafish starting from 0 dpf. Sample sizes: siblings n = 194, *vma21^Δ1/Δ1^* n = 74, *vma21^Δ14ins21/Δ14ins21^* n = 60, and *vma21^Δ1/Δ14ins21^* n = 60. As represented in the graph: WT siblings (dark blue), *vma21^Δ1/Δ1^* (light blue), *vma21^Δ14ins21/Δ14ins21^* (magenta), and vma21^Δ1/Δ14ins21^ (green). Significance at ****P < 0.0001 (Mantel–Cox test). (**H–J**) Free swim movement tracked over a ten-minute interval and (**K–M**) swim assay with the photoactive compound optovin to stimulate motor activity in 6 dpf larvae. Graphs from left to right show total time spent moving in seconds (s), total distance traveled in millimeters (mm), and average total velocity (mm/s) for each type of *vma21* mutant with their respective WT/HET sibling controls. n = 24/group with three replicates. Data represented as mean ± SEM. Significance at ****P < 0.0001 (unpaired two-tailed t test). As represented in the graphs: siblings (dark blue dots), *vma21^Δ1/Δ1^* (light blue dots), *vma21^Δ14ins21/Δ14ins21^* (magenta dots), and *vma21^Δ1/Δ14ins21^* (green dots). Source data are available online for this figure.

in *vma21*, we injected 1-cell stage embryos from a *vma21^Δ1^* HET incross with pTol2 (Ubbi: GFP-LC3-RFP-LC3ΔG) (Kaizuka et al, 2016), a fluorescent construct designed to evaluate autophagic flux, and imaged the larvae at 5 dpf. Compared to WT/HET clutchmates, *vma21* mutants had a higher GFP:RFP ratio (Fig. 4E–I), suggestive of lower autophagic flux.

## vma21 mutants have demonstrable liver pathology

While mutations in *vma21* are primarily associated with skeletal muscle disorders, several studies have reported hepatic injury in patients with mutations in V-ATPase assembly factors (Rujano et al, 2017; Jansen et al, 2016), and a recent study has linked this genetic deficiency with an autophagic hepatopathy wherein patients

experience elevated aminotransferases and low-density lipoprotein, hepatic steatosis, and mild cholestasis (Cannata Serio et al, 2020). To begin to investigate this, we performed routine histology using hematoxylin and eosin (H&E) staining. We observed considerable lipid deposition in the livers of *vma21* mutants as compared to WT controls (Fig. 5A,B), suggestive of hepatic steatosis and similar to that seen in patients. Based on general morphometric examination by light microscopy, liver size also appeared to be different. We subsequently confirmed, using the Crispant approach in a transgenic line marking the liver (*Tg(Fabp:mCherry)*), that liver size was significantly reduced in *vma21* crispants as compared to Cas9-only-injected controls (Fig. 5C–E), suggestive of liver dysfunction.

Next, we employed a BODIPY feeding assay to examine bile flux, as has previously been done to identify cholestasis in zebrafish

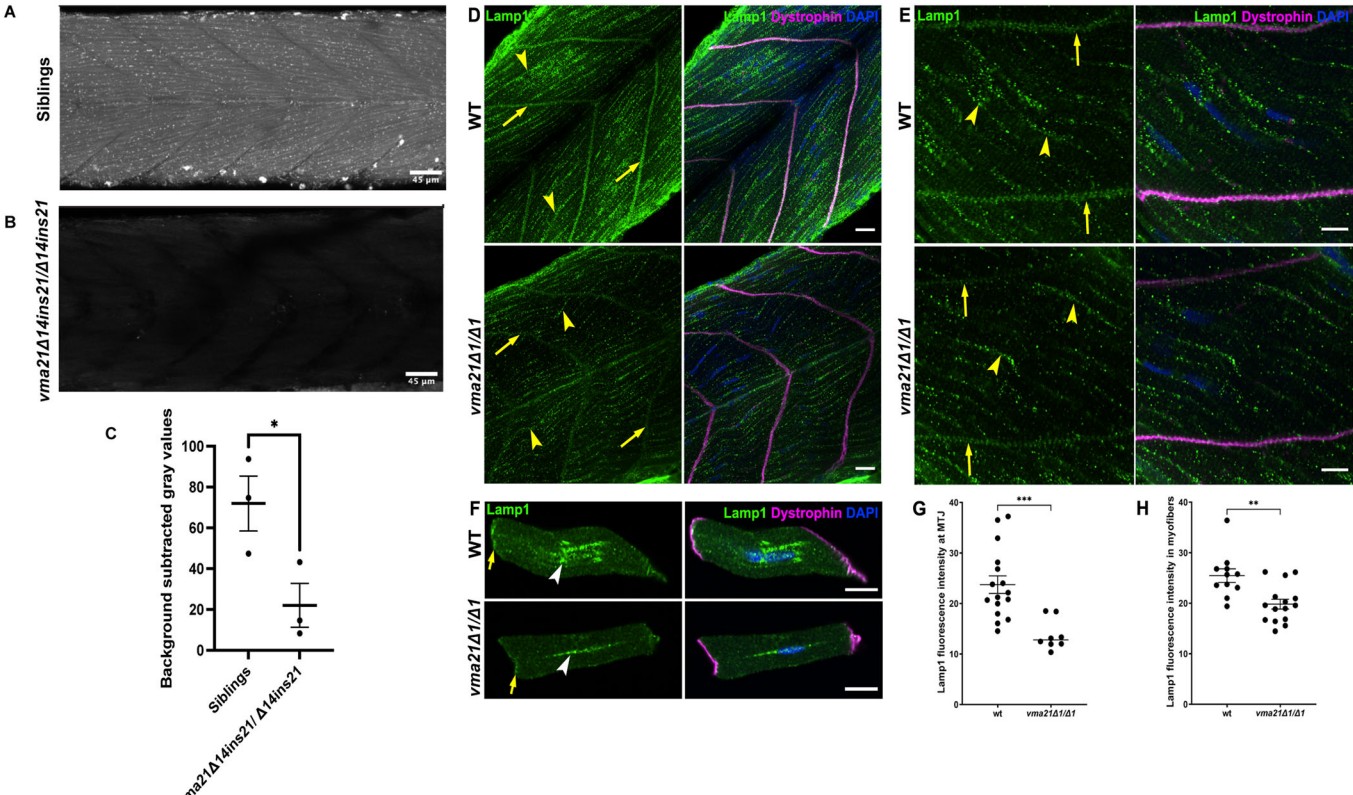

**Figure 3. Lysosomal acidification is impaired in *vma21* mutants.**

Representative images of (A) wild-type (WT)/heterozygous sibling control and (B) *vma21^Δ14ins21/Δ14ins21* zebrafish larvae at 5 days post fertilization (dpf) stained with LysoTracker Red, a marker of lysosomal acidification. Scale bars: 45 mm. (C) Bar graph showing quantification of LysoTracker Red staining. Each dot represents the average of gray values (technical replicate) and three biological replicates were performed. Data are shown as mean ± SEM. Significance at *$P < 0.05$ (unpaired two-tailed *t* test). (D) Z-stack and (E) single Z plane confocal micrographs of whole-mount zebrafish embryos at 5 dpf labeled with Lamp1 (green), Dystrophin (magenta), and DAPI (blue). Lamp1 (left panels in (D, E)) shows a punctate staining pattern and localizes in the myofibers (yellow arrowheads) and at the myotendinous junctions (MTJs) (yellow arrows). Lamp1 puncta are less bright in *vma21* mutants compared to WT. Scale bars: 20 mm (D) and 10 mm (E). (F) Single Z plane confocal micrographs of myofibers isolated from zebrafish embryos at 5 dpf and labeled with Lamp1 (green), Dystrophin (magenta), and DAPI (blue). Lamp1 puncta (left panels) are present throughout the myofibers, at the MTJs (yellow arrows), and in increased numbers around the nuclei (white arrowheads). Lamp1 puncta are less bright in *vma21* mutant myofibers compared to WT myofibers. Scale bar: 10 mm. (G) Quantification of Lamp1 fluorescence intensity at the MTJs shows significantly reduced levels of Lamp1 in *vma21* mutants. $n = 16$ WT and $n = 8$ *vma21* mutants. Data represented as mean ± SEM. Significance at ***$P < 0.001$ (unpaired two-tailed *t* test). (H) Quantification of Lamp1 fluorescence intensity in the entire myofiber shows significantly reduced levels of Lamp1 in *vma21* mutants. $n = 11$ WT and $n = 15$ *vma21* mutants. Data represented as mean ± SEM. Significance at **$P < 0.01$ (unpaired two-tailed *t* test). Source data are available online for this figure.

(Karolczak et al, 2023; Carten et al, 2011). Zebrafish larvae were fed food mixed with the fluorescent BODIPY analog from 4 to 6 dpf, imaged, and scored for fluorescence in the gallbladder, an indicator of normal bile flux. On average, based on three independent trials, ~8% of *vma21^Δ1/Δ1* mutants showed fluorescence in the gallbladder, which contrasts with WT controls (~76%) (Fig. 5F–H). These data suggest an impairment of bile flux in *vma21* mutant zebrafish, and imply an underlying cholestatic liver phenotype similar to that described in XMEA patients.

## Autophagy antagonist compounds improve survival and touch-evoked escape response but not impaired bile flux in vma21 mutant larvae

Given the aberrant autophagy observed with a loss of *VMA21* and the lack of therapeutic strategies for XMEA patients, we next performed a targeted drug modifier screen with autophagy inhibitory compounds. The Selleckchem autophagy drug library contains 1025 compounds from which 30 clinically tested autophagy antagonists were selected for evaluation. We first screened by mating *vma21^Δ14ins21* HET zebrafish and determining percent muscle birefringence, a measure of muscle organization (Smith et al, 2013), and survival in the resulting offspring. Based on Mendelian ratios in untreated clutches, it is expected that 25% of offspring (i.e., *vma21* mutants) will have abnormal birefringence. Drugs that lowered the percent affected birefringence to <18% were considered trending towards significance and those <12% considered a positive hit with the significance of $P < 0.05$. Using these outcome measures, we identified nine compounds (ROC-325, GSK2578215A, DC661, MRT68921 HCl, Lucanthone, CZC-25146, GNE7915, edaravone, and LY294002 administered at 2.5 μM) that both reduced the percentage of fish with abnormal birefringence (Fig. 6A) and prolonged survival of *vma21* mutants from a median lifespan of 10 to 14 dpf.

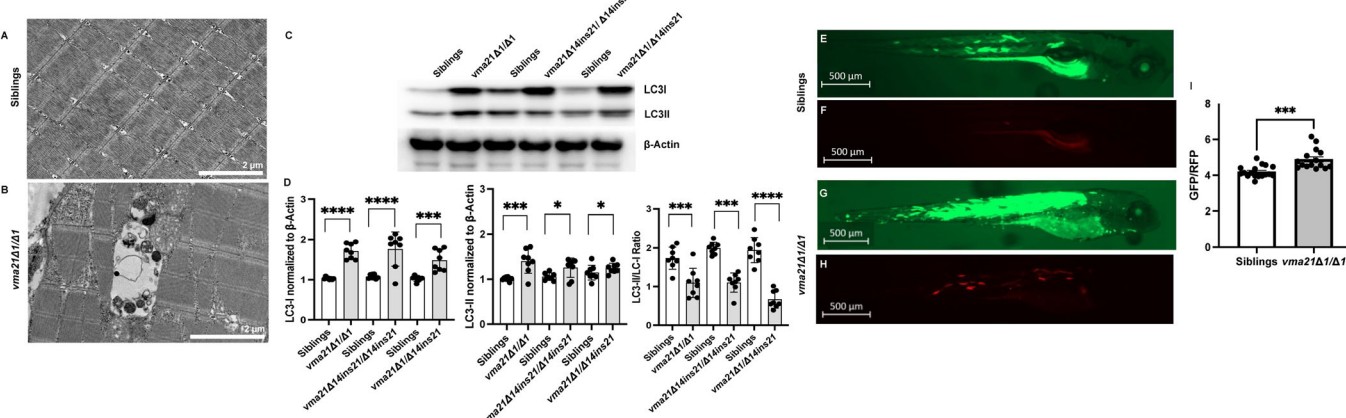

**Figure 4. Aberrant autophagy in *vma21* mutant zebrafish.**

Representative electron micrographs from (**A**) wild-type/heterozygous sibling control and (**B**) *vma21*$^{Δ1/Δ1}$ zebrafish larvae skeletal muscle at 6 days post fertilization (dpf), showing the presence of double-membrane vacuoles in *vma21* mutants but not controls. Scale bars: 2 mm. (**C**) Representative western blot of LC3I, LC3II, and b-actin (housekeeping protein) protein levels in the *vma21* 1 bp deletion (vma21Δ1/Δ1) and the 14 bp deletion with 21 bp insertion (vma21Δ14ins21/Δ14ins21) homozygous mutants as well as the compound heterozygous line (vma21Δ1/Δ 14ins21) compared to their respective wild-type/heterozygous sibling controls at 5 days post fertilization. LC3I protein is ~19 kDa, LC3II protein is approximately 17 kDa, and b-actin is approximately 42 kDa. Each lane represents $n = 10$ zebrafish (50 mg of total protein). Each western blot had a total of three biological replicates. (**D**) LC3I and LC3II protein levels were quantified by normalization to b-actin loading controls. The LC3II/LC3I ratio was also determined. For each sample, a total of $n = 30$ zebrafish were utilized. Data shown as mean ± SEM. Significance at *$P < 0.05$, **$P < 0.01$, ***$P < 0.001$, ****$P < 0.0001$ (one-way ANOVA). Representative fluorescent images of (**E, F**) wild-type/heterozygous sibling control and (**G, H**) *vma21*$^{Δ1/Δ1}$ zebrafish larvae in GFP and RFP channels at 5 dpf following injection of 1-cell stage embryos with the pTol2 (Ubbi: GFP-LC3-RFP-LC3ΔG) construct used to assess autophagic flux. Scale bar: 500 mm. (**I**) GFP/RFP ratio as a measure of autophagic flux in 5-dpf sibling control and *vma21*$^{Δ1/Δ1}$. $n = 15$–17/group. Data represented as mean ± SEM. Significance at ***$P < 0.001$ (unpaired two-tailed *t* test). Source data are available online for this figure.

We next evaluated the effect of long-term treatment of each of the nine autophagy antagonists identified in the initial screen on preventing lethality and extending the lifespan of *vma21* mutant zebrafish, with the experimental endpoint being either death or survival to 21 dpf (Fig. 6B). Untreated and vehicle-treated *vma21* mutant zebrafish had 100% lethality by ~10 dpf. In contrast, several autophagy antagonists showed significant improvements in survival, with edaravone having the greatest effect on lifespan extension (>60% survival by 21 dpf).

Next, using the touch-evoked escape response assay at 6 dpf, we evaluated the ability of the 30 autophagy antagonist drugs to rescue the impaired motor behavior phenotype observed in the *vma21* mutant fish. Vehicle-treated clutches (and negative hits) had, on average, 21% of the fish categorized as "low/no responders", consistent with the approximate expected ratio of homozygote mutant fish from a heterozygote in-cross. Of the drugs tested, edaravone- and LY294002-treated clutches showed the greatest reduction in "low/no responder" fish to 1.4% and 2.8%, respectively (Fig. 6C). These studies demonstrate the ability of autophagy inhibitors to improve functional outcomes in the *vma21*-deficient XMEA zebrafish.

In addition, the top two drug hits (edaravone and LY294002) were evaluated for their ability to correct the impaired bile flux liver phenotype in *vma21* mutants (Fig. 6D). Compared to DMSO-treated vma21$^{Δ1/Δ1}$ mutants, which showed ~20% BODIPY accumulation in the gallbladder, edaravone and LY294002-treated mutants showed ~20% and ~14%, respectively. Therefore, our findings suggest that autophagy antagonist compounds may be a potential therapeutic strategy to treat the muscle, but not the liver, phenotype observed in *vma21* mutant zebrafish.

Lastly, to evaluate the impact of the drugs on autophagy vis-à-vis *vma21* mutation, we looked by western blot at three markers of autophagy in untreated clutches and those treated with one of the top 9 compounds. Western immunoblots for LC3I/II showed significant reductions in *vma21* mutant zebrafish treated with ROC-325, DC661, MRT68921 HCl, CZC-25146, GNE7915, and LY294002 as compared to untreated mutant fish (Fig. EV2A,B). p62 (Sqstm1) (Fig. EV2C,D) and Becn1 (Atg6) (Fig. EV2E,F) were significantly reduced in the top 2 hits, edaravone and LY294002-treated zebrafish.

## Discussion

XMEA is a rare genetic muscle disease for which disease pathomechanisms are poorly understood and with no previously identified therapeutic strategies. A key limiting factor is the lack of validated vertebrate preclinical models of the disease. Thus, in this study, we created the first zebrafish model of a loss of *vma21* using CRISPR-Cas9 mutagenesis. The resulting mutant zebrafish phenocopy key features of the human disease. *vma21* mutants displayed overt phenotypic differences including abnormal morphology, impaired swim behavior, and reduced survival. They also had impaired lysosomal acidification and activity, as indicated by the absence and reduction of LysoTracker Red and Lamp1 staining, respectively. Moreover, *vma21* zebrafish displayed aberrant autophagy with electron-dense vacuoles, a hallmark of the disease, within the myofibers, increased LC3 protein levels, and reduced autophagic flux. We also show in mutant zebrafish the presence of hepatic steatosis, smaller liver size, and impaired bile flux, consistent with

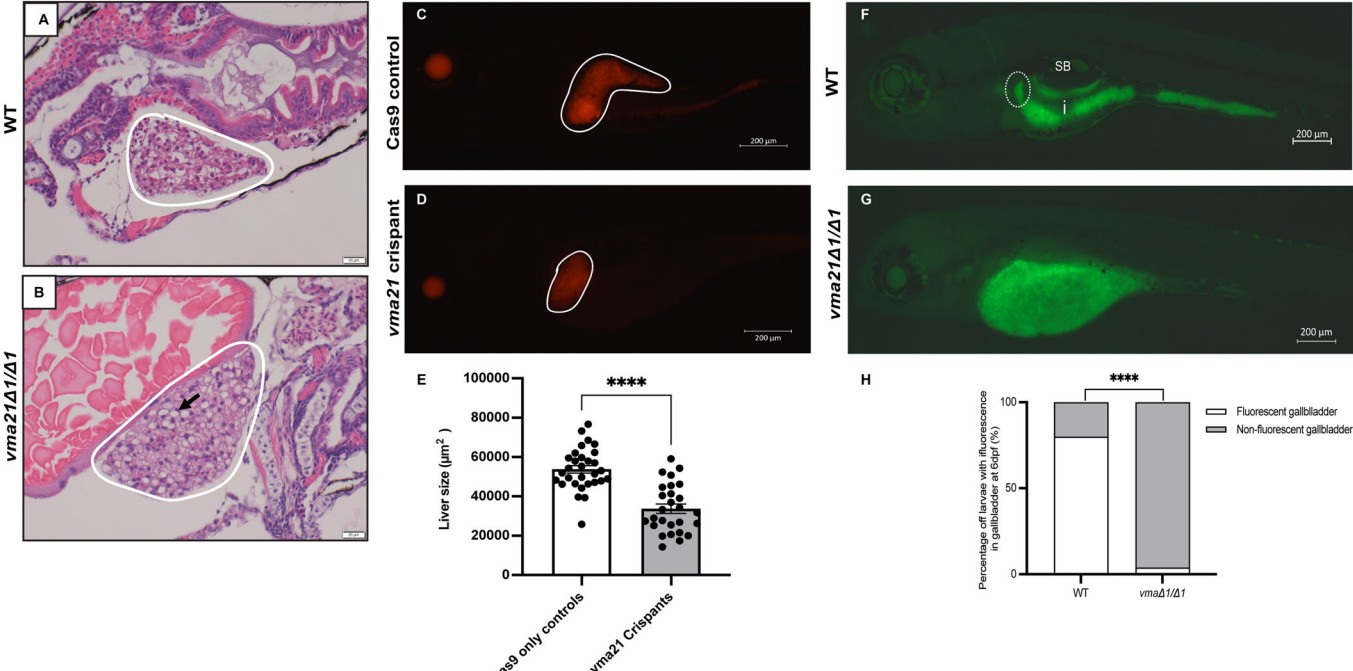

**Figure 5. *vma21* mutants have hepatic lipid accumulation, smaller liver size, and disrupted bile flux.**

Representative H&E staining of (A) WT and (B) *ma21^{Δ1/Δ1}* mutants at 6 dpf. The liver is outlined in white and the black arrow highlights a lipid droplet. *n* = 6 per group. Representative fluorescent images of the liver of (C) Cas9-only and (D) *vma21* crispants injected into the *Tg(Fabp:mCherry)* transgenic line at 5 dpf. The fluorescent red liver is outlined in white. (E) Quantification of liver size at 6 dpf. *n* = 27–30/group. Data represented as mean ± SEM. Significance at ****P < 0.0001 (unpaired two-tailed *t* test). Representative fluorescent images of (F) wild-type (WT) and (G) *vma21^{Δ1/Δ1}* mutants (homozygous for a 1 base pair (bp) deletion in *vma21*) at 6 days post fertilization (dpf) fed a BODIPY diet. The gallbladder is outline by the white dotted line. ×32 Magnification, scale bar: 200 mm. SB swim bladder, i intestine. (H) Quantification of the percentage of larvae with fluorescence in the gallbladder at 6 dpf. *n* = 26-29/group. Data represented as mean ± SEM. Significance at ****P < 0.0001 (Fisher's exact test). Source data are available online for this figure.

the reports of liver involvement in patients with *VMA21* mutations. Treatment with select autophagy antagonists improved survival and touch-evoked escape response outcomes but not the impaired bile flux in *vma21* mutant larvae. In total, these findings outline the first animal model of XMEA and support its use for investigating XMEA disease pathomechanisms and identifying potential therapies.

As a primary goal of this work is therapy development for XMEA, we used our model to perform a targeted investigation of drugs that modulate autophagy, mitophagy, and lysosomal biology. Chemicals that modulate these pathways were interrogated because of the previous reports of altered autophagy in XMEA patient cells (Ramachandran et al, 2013; Fernandes et al, 2020), and the known role of VMA21 in lysosomal function. Excitingly, we found that several autophagy antagonists could ameliorate aspects of the *vma21* zebrafish phenotype, and that two compounds in particular (edaravone and LY294002) improved the phenotype across multiple domains (birefringence, motor function, and survival). The fact that multiple autophagy modulators ameliorated aspects of the phenotype supports an important role for autophagy in the disease process and lends confidence to the validity and potential translatability of the findings.

Edaravone is a free radical scavenger with anti-oxidant and anti-inflammatory properties that has been tested in patients with amyotrophic lateral sclerosis (ALS) and with stroke (Abe et al, 2017; Tsukamoto et al, 2011) as well as in animal models of liver injury

(Tada et al, 2003; Ommati et al, 2021). Edaravone's exact mechanism of action in these diseases is not fully established, but preclinical studies support a role in reducing oxidative stress and modulating apoptosis (Guo et al, 2020). It can be orally dosed, has a favorable safety profile, and shows modest efficacy in slowing disease progression in ALS as well as a protective/preventative effect on hepatic injury. Thus, it has high potential translatability to patients with neurogenetic conditions including XMEA. LY294002 is a PI3 kinase inhibitor with action primarily against class I PI3 kinase (Vlahos et al, 1994). Exposure to LY294002 alters the AKT pathway and inhibits mTOR signaling, both of which can modulate autophagy among several other cellular processes (Blommaart et al, 1997). LY294002 has some clinical utility as an anti-cancer agent but given its side effect profile, it is likely not suitable for chronic use in patients. However, as there are several other compounds with better clinical safety that impact PI3 kinase/AKT signaling, the finding that LY294002 ameliorates the *vma21* zebrafish phenotype enables future evaluation of more suitable drugs with similar mechanisms of action.

In terms of therapy translation, a next logical step would be evaluation in a mammalian preclinical model and/or patient cell-derived disease avatar (such as iPS derived patient myotubes). At present, however, no such models have been established, though one study has reported characterization of myoblasts derived from an XMEA patient muscle biopsy (Fernandes et al, 2020). Future work is needed to develop and characterize new models, and then

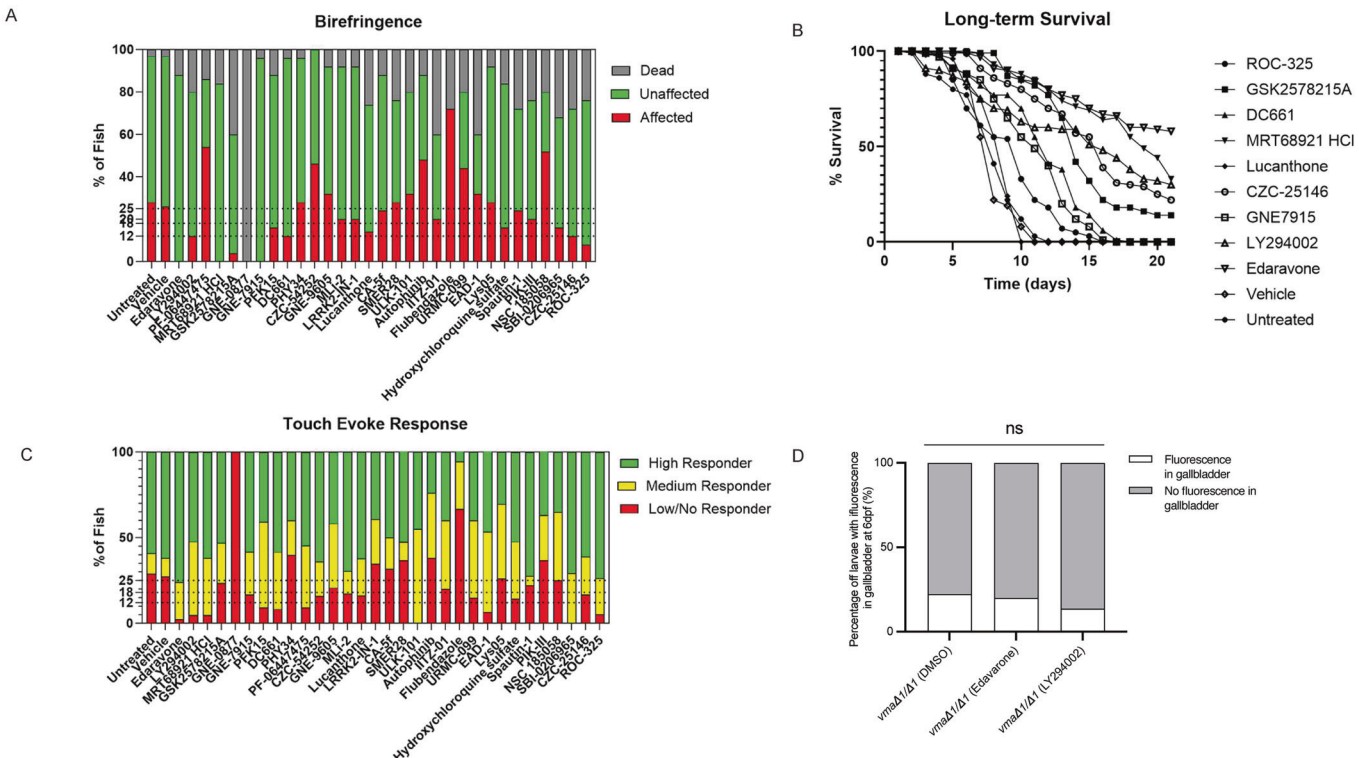

**Figure 6. Autophagy antagonist drug library screening identifies corrective molecules in the *vma21* mutant zebrafish.**

(A) Results from birefringence assessment of *vma21*^Δ14ins21^ heterozygote in-cross matings with zebrafish larvae incubated in autophagy library drug compounds. Dorsal muscle birefringence assessments of the drug-treated *vma21*^Δ14ins21/Δ14ins21^ homozygote zebrafish larvae offspring. Zebrafish were scored as dead (gray bars; no detectable heartbeat), affected (red bars; showing at least one area of poor birefringence), or unaffected (green bars; no birefringence). Dashed lines shown at 25%, 18%, and 12.5% target range for lead compound advancement. (B) Long-term survivability assessment of eight autophagy antagonists tested in the *vma21*^Δ14ins21/Δ14ins21^ homozygote mutant zebrafish advanced from the first pass short-term screen, LY294002 (positive control), and vehicle and untreated (negative controls). (C) Touch-evoke response measured with % of fish shown and groupings of high responders (green bars), medium responders (yellow bars), and low/non-responders (red bars). Compounds are administered into *vma21*^Δ14ins21/Δ14ins21^ homozygote mutant zebrafish at 1 dpf and drug water was changed every other day over a 21-day time period. The compounds include: ROC-325, GSK2578215A, DC661, MRT68921 HCl, Lucanthone, CZC-25146, GNE7915, edaravone, and LY294002 administered at 2.5 μM. Vehicle and untreated controls are also shown. One hundred (n = 100) *vma21*^Δ14ins21/Δ14ins21^ mutant fish were assessed over 3 independent trials. Compounds were later validated in the *vma21*^D1/D1^ mutant zebrafish. (D) Percentage of larvae with fluorescence in the gallbladder at 6 dpf following feeding with BODIPY as a measure of bile acid flux. n = 10–22/group. Data represented as mean ± SEM. Not significant (ns; P > 0.05, Fisher's exact test). Source data are available online for this figure.

to apply the therapeutic strategies we have identified. Additional future work in the fish will be aimed at dissecting the interplay between modulation of autophagy and the muscle pathology and weakness that results from *vma21* knockout, refining our initial drug "hits", and expanding our drug screen to a larger set of compounds. It is also important to note that some compounds (particularly wortmannin) worsened the *vma21* mutant phenotype, suggesting care must be taken in terms of choice of potential therapeutics to advance ultimately to patients. Lastly, the compounds identified as positive modulators of muscle phenotypes did not resolve the liver pathology, implying the potential for different disease pathomechanisms in muscle versus liver, and necessitating further study into the molecular events underlying the abnormalities that present in each organ system. There is also a need, therefore, to identify drugs which can improve alternations in the liver, and hopefully discovery of compound(s) that can improve both muscle and liver outcomes.

While the zebrafish recapitulates many features of human XMEA, it harbors important caveats both in general as a model system and in relation to XMEA. Zebrafish muscle is similar in

most regards to human muscle (Gibbs et al, 2013; Berger and Currie, 2012), but lacks (particularly during development) the interspersed patterning of type 1 and type 2 fibers seen in mammals (Tesoriero et al, 2023). Instead, fiber types are segregated in the fish, and often express individual paralogs of important muscle genes (for example, there are two paralogs of RYR1 (Hirata et al, 2007), one expressed in fast muscle and the other in slow muscle). Of note, zebrafish have a single VMA21 ortholog. More generally, while fish represent an outstanding model of many human disease types (Phillips and Westerfield, 2014), there are other differences, including, for example, that fish lack a pulmonary system similar to humans, and thus are not suitable for studying lung-related phenotypes.

With our *vma21* zebrafish, there are a few specific differences from human XMEA to note. First, the phenotype in our zebrafish is severe, and would be distinct from most patients with *VMA21* mutations, though does match what has been described in the most severe cases (Pegat et al, 2022; Blanco-Arias et al, 2023). This may be due to the fact that our model is a knockout while most patients likely express some low levels of VMA21. Second, while we

identified changes consistent with aberrant autophagy, we did not observe vacuoles with sarcolemmal features and multilayered basal membranes, a histopathologic abnormality associated with the majority of XMEA cases (60% in one study) (Fernández-Eulate et al, 2024). This may reflect species-specific differences or instead the age/stage at which we were able to evaluate the muscle. It is possible that older mutant fish may more clearly shows the characteristic vacuoles.

## Conclusion

In summary, we have established the first preclinical animal model of XMEA. We have determined that this model faithfully recapitulates most features of the human disease, and thus is ideally suited for establishing disease pathomechanisms and identifying therapies. As proof-of-concept, we performed a pilot drug screen using compounds that modulate autophagy and found several promising hits that warrant future study and have high potential for clinical translation.

## Methods

### Reagents and tools table

| Reagent/resource | Reference or source | Identifier or catalog number |
| --- | --- | --- |
| **Experimental models** | | |
| *vma21*$^{\Delta 1/\Delta 1}$ zebrafish | This study | N/A |
| vma21$^{\Delta 14ins21/\Delta 14ins21}$ zebrafish | This study | N/A |
| *Tg(Fabp:mCherry)* | This study | N/A |
| **Recombinant DNA** | | |
| pTol2 (Ubbi: GFP-LC3-RFP-LC3ΔG) | Kaizuka et al, 2016 | N/A |
| pcDNA3-GFP-LC3-RFP-LC3ΔG | Addgene | 168997 |
| **Antibodies** | | |
| Rabbit anti-VMA21 | Abcam | ab242099 |
| Rabbit anti-LC3B | Novus Biologicals | NB600-1384 |
| Rabbit anti-Becn1 | Santa Cruz Biotech | sc-11427 |
| Rabbit Anti-p62 (SQSTM1) | MBL Life Science | PM045 |
| Mouse anti-beta-actin | Abcam | ab8226 |
| Mouse anti-beta-actin | ThermoFisher Scientific | AM4302 |
| Goat Anti-Rabbit IgG (H + L)-HRP Conjugate | BioRad | 1706515 |
| Goat Anti-Mouse IgG (H + L)-HRP Conjugate | BioRad | 1706516 |
| Rabbit anti-Lamp1 | Abcam | ab24170 |
| Mouse anti-Dystrophin | Santa Cruz Biotechnology, Inc. | 7A10 |
| Alexa Fluor 488 goat anti-rabbit | ThermoFisher Scientific | A-11008 |
| Alexa Fluor 594 goat anti-mouse | ThermoFisher Scientific | A-11005 |
| **Oligonucleotides and other sequence-based reagents** | | |
| Zebrafish *vma21* gRNA | This study | See methods |
| *vma21* crispant gRNAs | This study | See methods |

| Reagent/resource | Reference or source | Identifier or catalog number |
| --- | --- | --- |
| HRM primers | This study | See methods |
| PCR sequencing primers | This study | See methods |
| Taqman probes | This study | See methods |
| **Chemicals, enzymes, and other reagents** | | |
| mMESSAGE mMACHINE™ SP6 Transcription Kit | Invitrogen™ | AM1340 |
| TaqMan™ Universal PCR Master Mix | Applied Biosystems™ | 4304437 |
| Cell lysis (RIPA) Buffer | Cell Signaling Technology | 9806 |
| Pierce™ BCA protein assay kit | ThermoFisher Scientific | 23225 |
| Clarity Max Western ECL Substrate | BioRad | 1705062 |
| Ethyl 3-aminobenzoate methanesulfonate salt (Tricaine) | MilliporeSigma | 5040 |
| Optovin analog 6b8 | ChemBridge | 5707191 |
| LysoTracker Red DND-99 | Invitrogen | L7528 |
| 1-phenyl-2-thiourea (PTU) | Sigma | P7629 |
| ProLong Gold Antifade Mountant with DAPI | ThermoFisher Scientific | P36935 |
| NEBuilder HiFi DNA Assembly Master Mix | New England Biolabs | E2621S |
| BODIPY™ FL C12 (4,4-Difluoro-5,7-Dimethyl-4-Bora-3a,4a-Diaza-s-Indacene-3-Dodecanoic Acid) | Invitrogen™ | D3822 |
| AP-100 Zebrafish feed | Ziegler | AP-100 |
| Autophagy Compound Library | Selleck Chemicals LLC | L2600 |
| LY294002 | Selleck Chemicals LLC | S1105 |
| Edaravone (MCI-186) | Selleck Chemicals LLC | S1326 |
| Dimethyl sulfoxide (DMSO) | MilliporeSigma | D2438 |
| Hematoxylin | Vector Laboratories | H-3401 |
| Eosin | Fisher Scientific | SE23-500D |
| **Software** | | |
| Chopchop | http://chopchop.cbu.uib.no/ | N/A |
| PRIMER-BLAST | https://www.ncbi.nlm.nih.gov/tools/primer-blast/ | N/A |
| GraphPad Prism version 9 | https://www.graphpad.com/ | N/A |
| Fiji ImageJ | https://imagej.net/software/fiji/ | N/A |
| NIS Elements software | Nikon Instruments Inc. | N/A |
| **Other** | | |
| LI-COR Odyssey DLx Imager | LI-COR Biosciences | N/A |
| Olympus BX43 | Olympus | N/A |
| Axio Zoom | Zeiss | N/A |
| ZebraBox | Viewpoint Behavior Technology | N/A |
| LSM710 confocal | Zeiss | N/A |

| Reagent/resource | Reference or source | Identifier or catalog number |
|---|---|---|
| A1R laser confocal | Nikon Instruments Inc. | N/A |
| RMB MT6000 ultramicrotome | Advanced Bioimaging Center, PGCRL, HSC | N/A |
| FEI Tecnai 20 transmission electron microscope | Advanced Bioimaging Center, PGCRL, HSC | N/A |
| SMZ1500 stereomicroscope | Nikon | N/A |

## Generation of *vma21* mutant zebrafish

*vma21* mutant zebrafish were generated in collaboration with the Zebrafish Genetics and Disease Models Core Facility at The Hospital for Sick Children (HSC, Toronto, Canada) using CRISPR-Cas9 mutagenesis. More specifically, 1-cell stage TU WT embryos were injected with 150 pg of Cas9 mRNA and a single guide RNA (gRNA, GATGATAACTCTACCCAT) at 10 pg that was designed to target exon 2, 116 bp downstream of the ATG start site, to produce knockout mutants. The program Chopchop (http://chopchop.cbu.uib.no/) was used to design the gRNA target sites, and gRNAs were synthesized by in vitro transcription (Hwang et al, 2013). The mMessage mMachine SP6 kit (Invitrogen) was used to synthesize the Cas9 mRNA via in vitro transcription. The software PRIMER-BLAST was used to design primers to evaluate gRNA cutting efficiency via high-resolution melt (HRM, forward 5'-CGGGTCTCTTGTCTCTGTGC-3' and reverse 3'-GCGAAA-TACCTTCAAAGAGAAGGG-5') analysis. To screen for germline transmission, sexually mature first-generation founder (F0) zebrafish were outcrossed to WTs and genomic DNA was extracted from individual F1 larvae for PCR amplification and sequencing to identify heterozygous carriers (*vma21* +/−) which were then in-crossed to generate F2 *vma21* −/− mutants. From the single gRNA, two *vma21* mutant lines were generated: (1) a 1 bp deletion (c.132delC, *vma21*$^{Δ1/Δ1}$) located in exon 2 resulting in a frameshift mutation without a premature stop codon, and (2) a 14 bp deletion with a 21 bp insertion (c.124_137del123_138insCTGAATATTA-CATCTGTATTT, vma21$^{Δ14ins21/Δ14ins21}$) also located in exon two, introducing a premature stop codon. Zebrafish were genotyped via HRM using TaqMan SNP genotyping assay according to the manufacturer's instructions (Applied Biosystems) and/or PCR amplification and sequencing (forward 5'-CGACAAGAAA-GAGGTGGGCT-3' and reverse 3'- AACGTTAGGTAAATA-CATGGCTGA-5'). vma21 disruption was also validated using western blot for protein levels.

To facilitate more rapid phenotypic investigations, we also developed *vma21* 'crispants' using the previously established F0 knockout method (Kroll et al, 2021). Briefly, three Alt-R™ CRISPR Custom Guide RNAs were ordered from IDT each targeting one of the three exons in the zebrafish vma21 gene (gRNA 1-AATTACGACAAGAAAGAGGT, gRNA2- CTGATGATAACTC-TACCCAT, gRNA 3-ATTGCTGGAGTAGCCCAAAG). One-cell stage zebrafish embryos were injected with 333 pg of each gRNA and 4700 pg of Cas9 protein (Alt-R® S.p. Cas9 Nuclease V3, IDT, Cat#1081058). Crispants were shown to recapitulate germline loss-of-function morphological phenotypes including non-inflated swim bladder and lack of pigmentation.

## Zebrafish husbandry and care

Zebrafish lines were bred and raised within the Zebrafish Facility at Peter Gilgan Center for Research and Learning (PGCRL, HSC, Toronto, CA) according to husbandry protocols guided by the Animal Care Committee. All adult fish were housed on a 14:10 h light/dark cycle and maintained at 28.5 °C. Embryos were raised to 5 dpf in a 28.5 °C incubator in embryo water (60 µg/ml Instant Ocean® sea salt with 0.1% methylene blue in system water, pH = 7.7). All experiments were conducted on larvae between the ages of 4–7 dpf with the approval of the HSC Animal Care Committee (Animal Use Protocol #676670).

## Western blots

Embryos (*n* = 10–30/group) were collected at 5 dpf and protein was extracted using Cell Lysis Buffer (Cell Signaling Tech, 9806) supplemented with a phosphatase and protease inhibitor cocktail. Briefly, samples were homogenized using a motorized pestle, stored on ice for ~20 min, centrifuged at 4 °C at 12,700 × *g* for 30 min, and subsequently the supernatant was collected. To quantify protein levels, a Pierce™ BCA protein assay kit (ThermoFisher Scientific) was used. 50 µg of protein was prepared with dithiothreitol and 4× loading dye. All samples were boiled for 5 min at 95 °C, before running on SDS-PAGE gels. Once the run was complete, the semi-dry transfer method was used to transfer the protein onto polyvinylidene fluoride (PVDF) membranes. Following the transfer, the membranes were blocked with 3% BSA for an hour at room temperature and then incubated with primary antibody at 4 °C overnight. The primary antibodies used included: anti-Vma21 (1:1000, Abcam ab242099), anti-LC3B (1:1000, Novus Biologicals NB600-1384), anti-Becn1 (1:1000, Cat# sc-11427; Santa Cruz Biotech); anti-p62 (1:1000, Cat# PM045; MBL Intl.) and anti-beta-actin (1:1000, ab8226, Abcam and 1:2000, clone AC-15; Cat# AM4302; ThermoFisher Scientific). All membranes were washed with 1X TBST between the incubations. The secondary antibodies used were anti-Rabbit-HRP (1/5000 BioRad 1706515) and anti-Mouse-HRP (1/5000 BioRad 1706516). Blots were imaged with either the Gel DocTM XER with Gel Documentation System (BioRad) using chemiluminescence with Clarity MaxTM ECL, BioRad or they were imaged on a LI-COR Odyssey DLx Imager (LI-COR Biosciences; Lincoln, NE), and densitometry quantification was assessed using the manufacturer's software. All western blots had three biological replicates. The quantification of the western blots was done through Fiji ImageJ and GraphPad Prism version 9.

## Bright-field and fluorescent imaging of larvae

Embryos were collected at various time points (between 4 and 6 dpf) for imaging. Embryos were anesthetized with 0.016% tricaine (MilliporeSigma: Cat# 5040) in embryo water and mounted in 3% methylcellulose in autoclaved system water on a single cavity depression slide. To image larvae, either an Olympus BX43 microscope under bright-field conditions and/or a Zeiss Axio Zoom macroscope under bright-field or fluorescent conditions were used.

## Morphological observation and body length measurements

Zebrafish larvae were raised to 5 dpf under standard conditions, anesthetized in tricaine, and mounted in 3% methylcellulose in a

single cavity depression slide. Bright-field images were taken at 10× of each individual larvae and after imaging, each larva was placed in a DNA lysis buffer for genotyping. All images were analyzed using Fiji ImageJ software. Briefly, body length measures were taken using the straight-line tool on ImageJ. We first performed a calibration on the image by measuring the distance of the image's scale bar and setting the scale of the image accordingly per pixel. Following calibration, measurements were taken using the straight-line tool and drawing a line from the head of the larvae to the end of the tail before the fin. Differences in pigmentation were noted by eye. The proportion of larva from each genotype with inflated swim bladders were also noted.

## Survival analysis

To analyze the survival of both *vma21*−/− zebrafish lines generated, respective *vma21*+/− zebrafish lines were in-crossed, and *vma21*−/− embryos from each clutch were separated at 4 dpf based on clear and consistent phenotypic differences (lack of pigmentation, smaller, and non-inflated swim bladder in *vma21*−/− zebrafish). These fish were screened for survival by monitoring their heartbeat visually under a light microscope. Embryos from each line were monitored daily from 4 to 14 dpf. Survival analysis was performed using the Kaplan–Meier method and the graph was generated using GraphPad Prism version 9. The survival curve was also assessed by the Mantel–Cox test.

## Touch-evoked escape response

*Vma21*$^{\Delta 14ins21}$ heterozygotes were bred, and the resulting clutches were analyzed for touch-evoked escape response using an established protocol (Smith et al, 2013; Sztal et al, 2016). At 6 dpf, larvae were placed individually into a plate and provoked with a gentle touch of the end of a pipette tip. Based on the level of motion after being touched, larvae were categorized into groups: "low/none responder → less than 500 µm of movement following probe stimulation", "medium responder → movement up to 5 cm following probe stimulation", and "high responder → movement greater than 5 cm following probe stimulation". Fish were assessed in a double-blinded manner (genotype and drug treatment).

## Swim assay

Swimming behavior was monitored in 5 dpf larvae using the ZebraBox platform (Viewpoint Behavior Technology, Lyon, France) as previously detailed (Smith et al, 2020, 2022). Briefly, embryos were individually plated into the wells of a 96-well plate in 150 µL of system water and treated with the photoactive optovin analog 6b8 (ChemBridge, Cat# 5707191, final concentration 10 µM) to stimulate motor behavior. After a 10 min incubation in the dark at 28.5 °C, the plate was placed into the Zebrabox. The protocol used consisted of: 30 s lights on, 1 min lights off, 30 s lights on, 1 min lights off, and 30 s lights on. To evaluate free-swimming behavior an alternative protocol without optovin stimulation was used consisting of 10 min under continuous light exposure. For all conditions, $n = 24$ larvae/group were used with three biological replicates for each experiment. Three quantitative parameters were measured including total time spent moving (seconds), distance traveled (millimeters), and average speed (millimeters/second). All the quantitative data was averaged out between the three trials and normalized.

## LysoTracker red staining

To begin to assess autophagy, zebrafish larvae were stained with LysoTracker Red (LysoTracker Red DND-99 Invitrogen, L7528), a red fluorescent dye that is cell-permeable and capable of staining acidic compartments including lysosomes, as previously detailed (He and Klionsky, 2010). Briefly, 24-h postfertilization embryos were placed in 1× PTU (1-phenyl-2-thiourea; Sigma, P7629) to prevent pigmentation. At 5 dpf, the larvae were transferred individually into the wells of a 96-well plate and LysoTracker Red was added to a final concentration of 10 µM. The plate was then incubated at 28.5 °C for 1 hr. Following the incubation, larvae were rinsed 3× with fresh embryo water. The whole larva was then mounted on a slide in glycerol and imaged on the LSM710 confocal microscope (Carl Zeiss AG). To quantify the LysoTracker intensity, Z-stack images were taken with the same microscope settings and three samples were used for siblings and mutants. Using Fiji ImageJ, gray values were measured for signals (for each sample, three measurements each for the fish muscle and background were taken and averaged). The background average was then subtracted from the average fish muscle lysotracker intensity, to obtain the final values.

## Fluorescence immunostaining and microscopy

Immunostaining of zebrafish whole-mount preparations was done as previously described (Smith et al, 2022) with the following modification: fixed embryos at 5 dpf were treated with pre-chilled acetone at −80 °C for 1 h. Embryos were incubated overnight, at 4 °C, with the following primary antibodies: rabbit anti-Lamp1 (1:200; ab24170; Abcam) and mouse anti-Dystrophin (1:20; 7A10; Santa Cruz Biotechnology, Inc.). The following secondary antibodies were used: Alexa Fluor 488 goat anti-rabbit (1:250; ThermoFisher Scientific) and Alexa Fluor 594 goat anti-mouse (1:250; ThermoFisher Scientific). Preparations were mounted with ProLong Gold Antifade Mountant with DAPI (ThermoFisher Scientific). Myofibers from 5 dpf zebrafish embryos were isolated and immunostained as previously described (Horstick et al, 2013; Zhao et al, 2019; Smith et al, 2020). The following primary antibodies were used: rabbit anti-Lamp1 (1:200; ab24170; abcam) and mouse anti-Dystrophin (1:100; 7A10; Santa Cruz Biotechnology, Inc.). The following secondary antibodies were used: Alexa Fluor 488 goat anti-rabbit (1:1000; ThermoFisher Scientific) and Alexa Fluor 594 goat anti-mouse (1:1000; ThermoFisher Scientific). Preparations were mounted with ProLong Gold Antifade Mountant with DAPI (ThermoFisher Scientific). Preparations were imaged with a Nikon A1R laser confocal microscope, using a 63×/1.4 oil objective. NIS Elements software (Nikon Instruments Inc., Melville, NY, USA) was used for image acquisition. Figures for publication were created in Adobe Photoshop (Adobe Inc., San Jose, CA, USA). Fluorescence intensity at the myotendinous junctions (MTJ) and in the myofibers was measured using ImageJ and plotted with GraphPad Prism.

## Electron microscopy

At 6 dpf, zebrafish larvae from a HET in-cross were separated according to their phenotype. Larvae were anesthetized in 0.016% (w/v) tricaine and fixed in approximately 1 mL of Karnovsky's

fixative for 2 h at room temperature. The solution was then replaced with fresh Karnovsky's fixative and left overnight at 4 °C. Samples were then submitted to the Advanced Bioimaging Center (PGCRL, The Hospital for Sick Children, Toronto) for processing. The larvae were rinsed in 0.1 M sodium cacodylate buffer, post-fixed for 90 min in 1% osmium tetroxide in buffer, dehydrated through an ethanol gradient series (50%, 70%, 90%, and 100%), and subsequently followed up by two 30 min changes in propylene oxide. Samples were then embedded in Quetol-Spurr resin and cured in a 60 °C oven overnight. An RMB MT6000 ultramicrotome was used to slice 90 nm sections and samples were then stained with uranyl acetate and lead citrate. Sections were imaged using an FEI Tecnai 20 transmission electron microscope.

## Dose-escalation ethanol (EtOH) treatment

As EtOH has previously been shown to induce whole-body autophagy in zebrafish, EtOH-exposed larvae were used as a positive control for the LC3 western blot (Wen et al, 2022; Varga et al, 2015). Briefly, 24 h postfertilization WT AB embryos were dechorionated and divided into control or EtOH exposure experimental groups, which were reared in either embryo water or 0.05, 0.5, or 1% EtOH. The water or alcohol was changed daily until 5 dpf. At 5 dpf, the larvae were frozen on dry ice for protein extraction and western blot.

## Autophagic flux assay

Autophagic flux was assessed using the pTol2 (Ubbi: GFP-LC3-RFP-LC3ΔG) construct that was synthesized by the Zebrafish Genetics and Disease Models Core facility (Sickkids, Toronto) as previously detailed (Kaizuka et al, 2016). To make the pTol2 (Ubbi: GFP-LC3-RFP-LC3ΔG) construct, the GFP-LC3-RFP-LC3ΔG open reading frame was amplified from pcDNA3-GFP-LC3-RFP-LC3ΔG (Addgene plasmid #168997) with primer pair F: GCCGCCACCATGGTGAGCAA; R: GTCACTATGGCGGCCGCATTGGCGGCCGTTACTAGTGGAT. The tol2 plasmid backbone and the ubiquitin promoter, was amplified with primer pairs tol2 F: ATACACAGCCAGTCTGCAGGT, tol2 R: AATGCGGCCGCCATAGTG; ubi F: CCTGCAGACTGGCTGTGTA-TACCAGCAAAGTTCTAGAATTTGTCG, ubi R: TTGCTCACCATG GTGGCGGCCTGTAAACAAATTCAAAG, respectively. Then, the three components mentioned above were assembled using NEBuilder HiFi DNA Assembly Master Mix (E2621S, New England Biolabs). The required 20 bp overlap sequences for Hifi assembly were underlined in the primer sequences above. Finally, the assembled construct was confirmed by sequencing. Following construct sequence confirmation, 1-cell stage zebrafish embryos from a *vma21*[Δ1] HET in-cross were injected with 30 pg of mRNA and 20 pg of plasmid. Embryos were then raised to 5 dpf, at which point they were anesthetized with tricaine, mounted in 3% methylcellulose and imaged under GFP and RFP conditions using the Zeiss Axiozoom. The GFP:RFP signal ratio, an estimate of autophagic flux, was measured using ImageJ by taking the fluorescence intensity of the GFP channel and dividing it by the fluorescence intensity of the RFP channel.

## Hematoxylin and eosin (H&E) staining

Zebrafish larvae from a *vma21*[Δ1] HET in-cross were raised to 6 dpf, fixed in 4% paraformaldehyde (PFA) for 24 h at 4 °C, and

submitted for paraffin embedding and sectioning at The Centre for Phenogenomics (TCP, Toronto, CA). Routine histological analysis was done via Mayer's Hematoxylin and Eosin (H&E) staining. Briefly, slides were deparaffinized with two changes of xylene, rehydrated in serial dilutions of ethanol, and subsequently stained with H&E. All images were taken on an Infinity1 camera (Lumenera Corp.) through an Olympus BX43 light microscope.

## Liver size

Liver size was investigated using the transgenic line *Tg(Fabp:m-Cherry)*. Briefly, 1-cell staged embryos from a *Tg(Fabp:mCherry)* outcross to WT ABs were injected with 3gRNAs with Cas9 protein to make *vma21* crispants or Cas9 only as a control. Zebrafish were raised to 5 dpf and imaged using the Zeiss Axio Zoom. The liver was outlined using a drawing tool on ImageJ and the area was then measured.

## Bile flux assay

In vivo analysis of cholesterol metabolism was performed as previously described (Chen et al, 2018; Dai et al, 2015; Ma et al, 2019). Briefly, BODIPY-C12 (ThermoFisher, catalog #: D3822) was reconstituted in 100% ethanol. 1 mL of this solution was then combined with 1 mL of chloroform and added to 0.5 g of the AP-100 Zebrafish feed (Ziegler, 50-μm size) in a ceramic dish loosely covered with aluminum foil and placed under a fume hood for 3 h to allow the chloroform to evaporate. Zebrafish larvae from a *vma21*[Δ1] heterozygous in-cross were then fed the AP-100/BODIPY feed (3.75 mg per 25 fish) from the afternoon of 4 dpf until the morning of 6 dpf with water and feed being replaced daily. At 6 dpf, larvae were rinsed with embryo water to remove excess feed, anesthetized with tricaine, and imaged on a Zeiss Axio Zoom macroscope. Fluorescence in the intestine (to confirm feeding) and gallbladder were scored for each larva.

## Autophagy drug library and drug compound zebrafish evaluation

The Selleckchem autophagy inhibitor drug compound library (Selleck Chemicals LLC; Houston, TX; Cat# L2600) was obtained commercially, and 30 validated and clinically tested autophagy inhibitors were tested in short-term drug screening in the *vma21*[Δ14ins21] mutant zebrafish. The PI3K inhibitor LY294002 (Selleck Chemicals; Cat# S1105) was used as a positive control for all experiments along with edaravone (MCI-186; Selleck Chemicals; Cat# S1326) in later drug evaluation experiments. All drug compounds were diluted in either pure fish water (hydrophilic compounds) or vehicle (hydrophobic compounds) consisting of 0.01% DMSO (D2438; MilliporeSigma; Burlington, MA) dissolved in fish water. Drug testing was performed on larvae from *vma21*[Δ14ins21] heterozygous in-cross. Drug treatment began when larvae were 1 dpf, and continued every other day until completion of the experiment (water changes occurred on the alternate days in between drug treatments). The resulting offspring ($n = 25$ per treatment) were then placed in the drug pools treated with either vehicle, positive control (2.5 μM LY294002) or 2.5 μM of the experimental autophagy library drug compound in 24-well dishes. Expected Mendelian ratios of 25% WT, 50% heterozygotes, 25%

homozygote mutants were assessed for every experimental treatment and genotypes were later confirmed via Sanger PCR genomic DNA sequencing. Drugs that showed toxicity in the short-term assessments were later re-evaluated at lower drug concentrations (0.1 nM, 10 nM, 100 nM, and 1 μM) for toxicity effects. However, no drug compound showing toxicity at the initial drug concentration of 2.5 μM showed therapeutic efficacy at the lower doses. Assessments of skeletal muscle birefringence and total survival were used as primary endpoints with touch-evoked escape response as a secondary biomarker as previously described. Experiments were concluded at 6 dpf, and unfertilized eggs were removed from the experimental counts.

Long-term (up to 21 dpf) testing was conducted with survivability as the primary endpoint. For the long-term assessments, the $vma21^{\Delta14ins21}$ homozygote mutant zebrafish were separated from the WT/heterozygote sibling mixed cohorts and later confirmed via Sanger PCR genomic DNA sequencing. At 7 dpf the zebrafish were moved from 24-well dishes to 100 mL isolation tanks with the corresponding drugs. Fish were monitored daily for death events which were tallied per tank. Fish were assessed in a double-blinded manner (genotype and drug treatment cohorts). Three independent trials being conducted with a minimum of $n = 100$ larvae for the short-term screening and $n = 30$ larvae for the long-term assessments. Drug library short-term screening was conducted $vma21^{\Delta14ins21}$ homozygote zebrafish and long-term testing was performed separately in both the conducted $vma21^{\Delta14ins21}$ and $vma21^{\Delta1}$ zebrafish strains with confirmatory results observed in both mutant lines.

### Skeletal muscle birefringence

In all, 4–6 dpf zebrafish larvae were anesthetized with 0.016% (w/v) tricaine and mounted on a single cavity depression slide with 3% methylcellulose. To assess muscle integrity, larvae were imaged using a Nikon SMZ1500 stereomicroscope (Nikon; Tokyo, Japan) fitted with two polarizing filters. Birefringence intensity was quantified using ImageJ software by measuring pixel brightness (mean gray value) along the larvae's ventral and dorsal plot profiles.

### Statistical analysis

All statistical analyses were conducted using GraphPad Prism (version 9; GraphPad Software, LLC., Boston, MA 02110, USA). A power analysis was conducted to determine sample size, and most analyses were conducted in a blinded manner with genotyping conducted following the experimental investigation. A Student's t test was performed to investigate differences between control (WT alone or WT and HET siblings combined or Cas9 only) and $vma21$ mutant zebrafish including for western blot, swim assay, LysoTracker Red staining, immunostaining, autophagic flux, and liver analyses. A one-way ANOVA was used to determine differences between all three genotypes followed up by Tukey's post-hoc or Dunnett's test for pairwise comparisons. For statistical analysis of proportions including proportion of inflated swim bladders and the bile flux assay with BODIPY, Fisher's exact test was used. Significance is reported at $P < 0.05$ and all values are reported as mean ± SEM. Exact $n$ and $P$ values can be found in Appendix 1.

## Data availability

This study includes no data deposited in external repositories.

The source data of this paper are collected in the following database record: biostudies:S-SCDT-10_1038-S44321-025-00204-8.

## Peer review information

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

## Acknowledgements

This work was supported by grants from CIHR/RDMM. The funders had no role in the study design, the collection and analysis of data or the preparation of the manuscript. In addition, we would like to thank the Zebrafish Genetics and Disease Model Core facility at the Hospital for Sick Children for their help in generating our zebrafish mutant model and the construct for assessing autophagic flux.

## Author contributions

**Lily Huang**: Data curation; Formal analysis; Validation; Investigation; Visualization; Methodology; Writing—original draft. **Rebecca Simonian**: Data curation; Formal analysis; Validation; Investigation; Visualization; Methodology; Writing—original draft; Writing—review and editing. **Michael A Lopez**: Data curation; Formal analysis; Investigation; Methodology. **Muthukumar Karuppasamy**: Data curation; Formal analysis; Investigation; Methodology. **Veronica M Sanders**: Data curation; Formal analysis; Investigation; Methodology. **Katherine G English**: Data curation; Formal analysis; Investigation; Methodology. **Lacramioara Fabian**: Data curation; Formal analysis; Investigation; Methodology. **Matthew S Alexander**: Conceptualization; Resources; Supervision; Funding acquisition; Writing—original draft; Project administration; Writing—review and editing. **James J Dowling**: Conceptualization; Resources; Supervision; Funding acquisition; Validation; Visualization; Writing—original draft; Project administration; Writing—review and editing.

Source data underlying figure panels in this paper may have individual authorship assigned. Where available, figure panel/source data authorship is listed in the following database record: biostudies:S-SCDT-10_1038-S44321-025-00204-8.

## Disclosure and competing interests statement

The authors declare no competing interests.

# Expanded View Figures

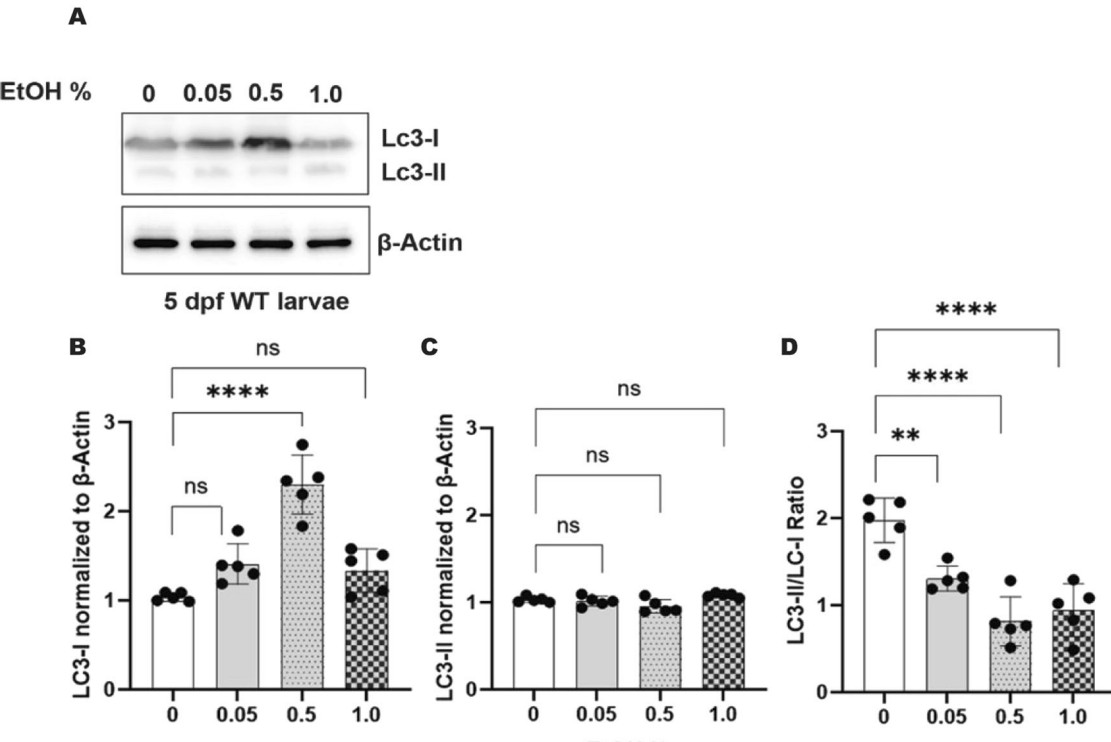

**Figure EV1.  Ethanol exposures increase whole-body autophagy in zebrafish.**

(A) Western immunoblots for LC3I, LC3II, and β-actin performed on ethanol (EtOH)-treated (0, 0.05, 0.5, and 1% EtOH) zebrafish. Densitometry quantification showing the expression for (B) LC3I, (C) LC3II, and (D) LC3II/LC3I ratio normalized to β-actin control. One-way ANOVA are shown with *P* values indicated as the following: *$P < 0.05$; **$P < 0.01$; ***$P < 0.005$; ****$P < 0.001$. Each western blot had a total of five biological replicates. For each sample, $n = 30$ zebrafish were utilized. Data represented as mean ± SEM. Source data are available online for this figure.

A.

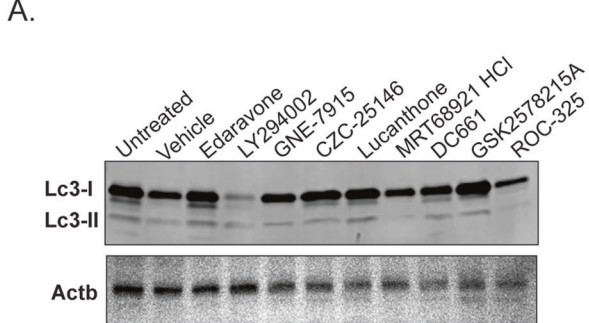

B.

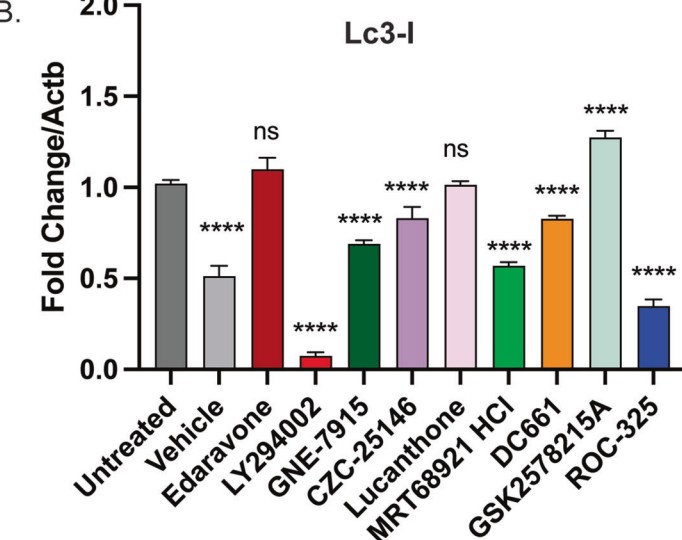

C.

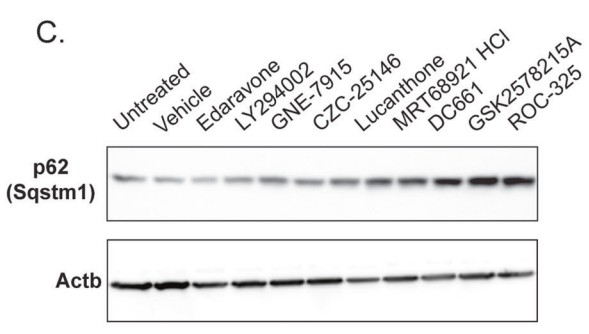

D.

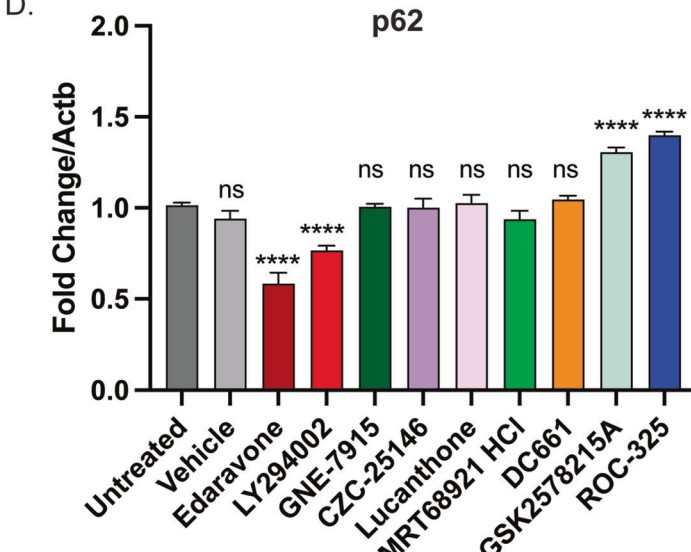

E.

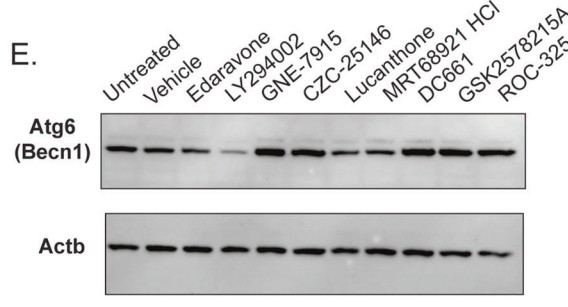

F.

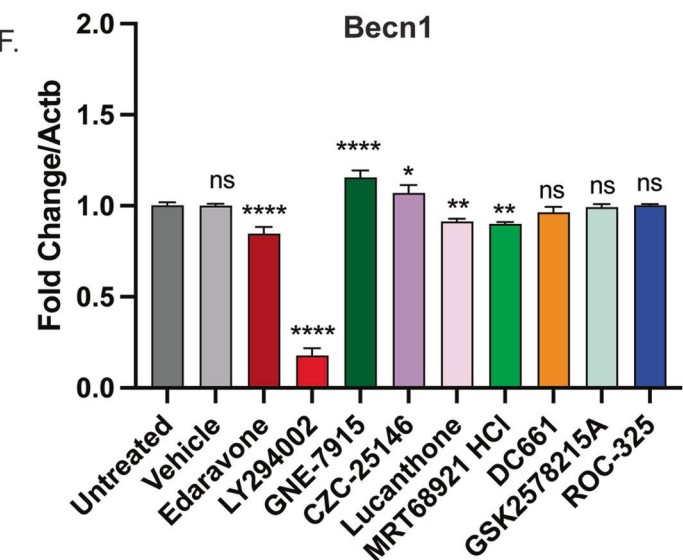

◀ **Figure EV2.  Autophagy markers are reduced in drug-treated *vma21* mutant zebrafish.**

Western immunoblots of drug-treated zebrafish as well as densitometry quantification showing the expression in drug-treated *vma21*Δ$^{14ins21/Δ14ins21}$ mutant homozygote mutant fish cohorts for (**A**, **B**) Lc3-I/II, (**C**, **D**) p62 (Sqstm1), and (**E**, **F**) Becn1 (Atg6) normalized to β-actin (Actb) loading controls. Nine drug compounds (Edaravone, LY294002, GNE7915, CZC-25146, Lucanthone, MRT68921 HCl, DC661, GSK2578215A, ROC-325) along with untreated and vehicle controls were evaluated. One-way ANOVA are shown with *P* values indicated as the following: *$P < 0.05$; **$P < 0.01$; ***$P < 0.005$; ****$P < 0.001$. Each western blot had a total of three biological replicates. For each sample, $n = 30$ zebrafish were utilized. Data represented as mean ± SEM. Source data are available online for this figure.

    