## [Peer Review File · EMBO Molecular Medicine]

X-linked myopathy with excessive autophagy: characterization and therapy testing in zebrafish model

Lily Huang, Rebecca Simonian, Michael Lopez, Muthukumar Karuppasamy, Veronica Sanders, Katherine English, Lacramioara Fabian, Matthew Alexander, and James Dowling

Corresponding author(s): James Dowling (james.dowling@sickkids.ca), Matthew Alexander (matthewalexander@uabmc.edu)

Review Timeline:

Submission Date:	12th Jan 23
Editorial Decision:	1st Feb 23
Revision Received:	16th Sep 24
Editorial Decision:	1st Oct 24
Revision Received:	3rd Feb 25
Accepted:	10th Feb 25

Editor: Zeljko Durdevic

Transaction Report:

1st Feb 2023

Dear Dr. Dowling,

Thank you for the submission of your manuscript to EMBO Molecular Medicine. We have now received feedback from the three reviewers who agreed to evaluate your manuscript. As you will see from the reports below, referees recognize potential interest of the study, but also raise serious and overlapping concerns particularly regarding the limited characterization of the zebrafish model and the lack of mechanistic insight. We would like to invite revision of the current manuscript with the focus on detailed characterization of the zebrafish model and providing mechanistic insights into disease pathology and mode of action of potential therapeutics. If you would like to discuss further the points raised by the referees, I am available to do so via email or video. Let me know if you are interested in this option.

We would welcome the submission of a revised version within three months for further consideration. Please let us know if you require longer to complete the revision.

I look forward to receiving your revised manuscript.

Yours sincerely,

Zeljko Durdevic

We require:

- 1) A .docx formatted version of the manuscript text (including legends for main figures, EV figures and tables). Please make sure that the changes are highlighted to be clearly visible.
- 2) Individual production quality figure files as .eps, .tif, .jpg (one file per figure). For guidance, download the 'Figure Guide PDF': (<https://www.embopress.org/page/journal/17574684/authorguide#figureformat>).
- 3) A .docx formatted letter INCLUDING the reviewers' reports and your detailed point-by-point responses to their comments. As part of the EMBO Press transparent editorial process, the point-by-point response is part of the Review Process File (RPF), which will be published alongside your paper.
- 4) A complete author checklist, which you can download from our author guidelines (<https://www.embopress.org/page/journal/17574684/authorguide#submissionofrevisions>). Please insert information in the checklist that is also reflected in the manuscript. The completed author checklist will also be part of the RPF.
- 5) Please note that all corresponding authors are required to supply an ORCID ID for their name upon submission of a revised manuscript.

6) It is mandatory to include a 'Data Availability' section after the Materials and Methods. Before submitting your revision, primary datasets produced in this study need to be deposited in an appropriate public database, and the accession numbers and database listed under 'Data Availability'. Please remember to provide a reviewer password if the datasets are not yet public (see <https://www.embopress.org/page/journal/17574684/authorguide#dataavailability>).

.

- the medical issue you are addressing,

- the results obtained and

- their clinical impact.

13) Author contributions: You will be asked to provide CRediT (Contributor Role Taxonomy) terms in the submission system. These replace a narrative author contribution section in the manuscript.

14) A Conflict of Interest statement should be provided in the main text.

Please note: When submitting your revision you will be prompted to enter your funding and payment information. This will allow Wiley to send you a quote for the article processing charge (APC) in case of acceptance. This quote takes into account any reduction or fee waivers that you may be eligible for. Authors do not need to pay any fees before their manuscript is accepted and transferred to the publisher.

EMBO Press participates in many Publish and Read agreements that allow authors to publish Open Access with reduced/no publication charges. Check your eligibility: <https://authorservices.wiley.com/author-resources/Journal-Authors/open-access/affiliation-policies-payments/index.html>

**** Reviewer's comments ****

Referee #1 (Comments on Novelty/Model System for Author):

At this stage, it is unclear how appropriate this zebrafish model is to adequately study human XMEA. Further, no new mechanistic insights are provided.

Referee #1 (Remarks for Author):

This manuscript characterizes a new gene-targeted zebrafish model to study human XMEA. While the creation of a model system to study human XMEA is highly warranted, this report falls short of describing novel insights into disease mechanisms and related therapeutics. Comments are below.

General comments:

- 1) This XMEA zebrafish model does not phenocopy clinical XMEA in terms of mortality, raising concerns over the validity of this model. The authors must better describe the mechanism of premature death, and clearly articulate the appropriateness of this system to model human XMEA. Perhaps a mouse model would have been more appropriate, where one can better study clinically-relevant parameters such as muscle function and ambulation.
- 2) The authors report this gene-targeted zebrafish model follows Mendelian inheritance, but provide no proof of such. In particular, the authors weigh heavily on such genetics to assay therapeutic efficacy of autophagy-targeting molecules.
- 3) The major theme of this report is the characterization of a new zebrafish model, but provides no novel insights into disease mechanisms.
- 4) Survival is not a major clinical endpoint in XMEA, questioning the relevance of this readout as a surrogate of clinical efficacy of Edaravone and other autophagy inhibitors. No attention was given to the discordance in responders vs non-responders to the touch response.

Specific comments:

- 1) No evidence of off-target gene editing effects were reported. Gene-corrected 'healthy' control is more appropriate than wild-type zebrafish.
- 2) The discussion is far too narrative.

Referee #2 (Remarks for Author):

The paper by Huang et al reports on a zebrafish model for Vma21 deficiency. Mutations in Vma21 cause several genetic

diseases, including X-linked myopathy with excessive autophagy (XMEA), for which treatments are lacking. Vma21 mutant fish are featured by short survival, muscle weakness, lysosomal acidification defects and excessive autophagy in muscle. The authors also identify several autophagy antagonists that alleviate the touch evoked escape response and can extend survival. Altogether, the paper is of clinical relevance and therefore interesting. But it needs substantial revision. Most importantly, it will be necessary to characterize the effect of the drugs in more detail. Currently, there is not much information on their mode of action. This is particularly important as the different drugs target additional biological processes and pathways besides autophagy. I suggest to study fish morphology, swim behaviour, autophagy (LC3I/II levels) and Lysotracker staining for all 8 compounds that were identified in the initial screen (ROC- 325, GSK2578215A, DC661, MRT68921 HCl, Lucanthone, CZC-25146, GNE7915, edaravone, and LY2940020).

Additional points:

1. Western Blot in Fig. 5c does not look really convincing. Also, why are the bands at different heights?
2. It is not clear when the liver phenotype was studied. This should be looked at during late time points as it might take time point for the liver phenotype to develop.
3. Vma21 patients with liver disease also have glycosylation defects. If possible in fish, this should also be studied.

Referee #3 (Comments on Novelty/Model System for Author):

Pathological characterization of the zebrafish model is not enough while the disease is an entity mainly defined by pathological features in human patients (see the "Remarks to be sent to the author").

Referee #3 (Remarks for Author):

Huang et al report a novel zebrafish model of X-linked myopathy with excessive autophagy (XMEA), which is actually the first animal model of the disease, and the authors claim that their model replicates most features of the human disease. Furthermore, the authors identified that edaravone and LY294002 improve the phenotype.

This is an important paper but I have two major concerns.

1) I am afraid that the pathological characterization of the phenotype is insufficient considering that XMEA is an entity mainly defined by pathological features. In human, XMEA is pathologically characterized by: 1) autophagic vacuoles with unique sarcolemmal features (AVSF), 2) multilayered basal lamina with exocytosed materials in between the layers, 3) the deposition of calcium and C5b-9 on the sarcolemma; while the authors have shown only the presence of autophagic vacuoles. Pathological features of the model zebrafish should be characterized in more details focusing on the abovementioned features.

2) I wonder what would be the mechanism of edaravone and LY294002 to improve the phenotype. The authors use the term "autophagy antagonist" but I wonder if the phenotype improvement is truly due to the inhibition of autophagy. This should be assessed in more details for example by knocking down essential genes in autophagy in the zebrafish model. In the same line, the authors may want to evaluate acidification status in the treated zebrafish.

We thank the reviewers and the editor for the thorough evaluation, and for the opportunity to submit a resubmitted version of our manuscript. In response to reviewer critiques, we have performed several additional experiments. New data include the following: (1) evaluation of the liver phenotype of *vma21* mutants, with a demonstration of impaired bile flux consistent with a cholestatic phenotype (new Figure 5); (2) additional evaluation of drug treatments, with a thorough assessment of treatment effect on phenotype (birefringence, motor function, and survival) and autophagic markers (new Figure EV 1); (3) additional evaluation of lysosomes and autophagy in *vma21* mutants.

Below, we provide a point-by-point response to the individual reviewers comments.

***** Reviewer's comments *****

Referee #1 (Comments on Novelty/Model System for Author):

At this stage, it is unclear how appropriate this zebrafish model is to adequately study human XMEA. Further, no new mechanistic insights are provided.

We put forward that our model recapitulates key features of the human disease that make it suitable for studying XMEA. (1) It harbors loss of expression mutations in *vma21*, consistent with the known mutational mechanism of XMEA (i.e. loss of expression/function of VMA21). (2) It has significant motor impairment (similar to patients with severe disease) and cholestatic liver changes. (3) There is evidence of aberrant autophagy on muscle histopathology and via molecular analyses. (4) There is evidence of aberrant lysosome acidification, the major cellular abnormality predicted to result from *vma21* mutation.

The goals of the study were to establish a pre-clinical model of XMEA (as none currently exist) and to utilize it to test potential therapeutic strategies. We believe we have succeeded on both fronts and produced a study of importance to the field.

Referee #1 (Remarks for Author):

This manuscript characterizes a new gene-targeted zebrafish model to study human XMEA. While the creation of a model system to study human XMEA is highly warranted, this report falls short of describing novel insights into disease mechanisms and related therapeutics. Comments are below.

General comments:

1) This XMEA zebrafish model does not phenocopy clinical XMEA in terms of mortality, raising concerns over the validity of this model. The authors must better describe the mechanism of premature death, and clearly articulate the appropriateness of this system to model human XMEA. Perhaps a mouse model would have been more appropriate, where one can better study clinically-relevant parameters such as muscle function and ambulation.

Premature death has been reported in patients with severe XMEA, so our data is not inconsistent human clinical presentations. In animal models in general it is often difficult to enumerate the exact cause of death (including mice as well). Zebrafish with impaired swimming are unable to adequately access food and have reduced ventilation (as oxygenation is supported by swimming). These are the likely causes of death (reduced feeding and/or impaired respiration) in our zebrafish.

Zebrafish have a proven track record as a model system for studying neuromuscular disorders and for identifying therapeutic approaches. Mice are also an excellent model system, and a mouse model of XMEA (provided it is viable and manifests suitable phenotype) will also be useful for studying XMEA. However, this is beyond the scope of this study, and there is certainly merit to both the zebrafish and the murine model systems.

2) The authors report this gene-targeted zebrafish model follows Mendelian inheritance, but provide no proof of such. In particular, the authors weigh heavily on such genetics to assay therapeutic efficacy of autophagy-targeting molecules.

There are multiple lines of evidence that support a Mendelian inheritance. Crosses of heterozygous fish yield the anticipated genotypes (+/+, *vma21*/+, *vma21*/*vma21*) at the expected ratio (1:2:1). Only *vma21*/*vma21* fish have a phenotype (conforming to recessive Mendelian inheritance). Multiple lines were created with mutations in *vma21*. Each yield the same *vma21*/*vma21* phenotype. Compound heterozygous fish also have the same phenotype.

3) The major theme of this report is the characterization of a new zebrafish model, but provides no novel insights into disease mechanisms.

The major goals of this study were to establish and characterize the first pre-clinical model of XMEA, and to use this model to test potential therapeutic strategies. We believe we have succeeded in both of these goals. Our model is well set for future mechanistic studies, but these were not the major foci of the present report.

4) Survival is not a major clinical endpoint in XMEA, questioning the relevance of this readout as a surrogate of clinical efficacy of Edaravone and other autophagy inhibitors. No attention was given to the discordance in responders vs non-responders to the touch response.

We studied three outcomes in relation to different treatments, all of which are clinically relevant. The first was birefringence, which is a sensitive measure of muscle structural integrity. The second was survival, and impaired survival has been reported in severe cases of XMEA (see Blanco-Arias et al., 2023, Neuromuscular Disorders). The third was touch evoked escape response, a well-established motor functional measure that is a surrogate for muscle force generation.

For touch response, our screen was for reduction in the % of embryos with no response. We did this on a population basis, and did not focus on the individual variance (i.e. no, low, high responder) within treatment groups. It is an interest point related to responder vs non-responder within effective treatments, though not one we have pursued at this point. We appreciate the question, and plan to look further at this in future studies.

Specific comments:

1) No evidence of off-target gene editing effects were reported. Gene-corrected 'healthy' control is more appropriate than wild-type zebrafish.

We appreciate the point. However, we did not observe any differences between +/+ and +/- fish from our studies groups and fish that have not been exposed to gene editing machinery. Also, "wild type" is standard in the field. We therefore have kept this terminology.

2) The discussion is far too narrative.

We have edited the discussion to include less exposition.

Referee #2 (Remarks for Author):

The paper by Huang et al reports on a zebrafish model for Vma21 deficiency. Mutations in Vma21 cause several genetic diseases, including X-linked myopathy with excessive autophagy (XMEA), for which treatments are lacking. Vma21 mutant fish are featured by short survival, muscle weakness, lysosomal acidification defects and excessive autophagy in muscle. The authors also identify several autophagy antagonists that alleviate the touch evoked escape response and can extend survival.

Altogether, the paper is of clinical relevance and therefore interesting. But it needs substantial revision. Most importantly, it will be necessary to characterize the effect of the drugs in more detail. Currently, there is not much information on their mode of action. This is particularly important as the different drugs target additional biological processes and pathways besides autophagy. I suggest to study fish morphology, swim behaviour, autophagy (LC3I/II levels) and Lysotracker staining for all 8 compounds that were identified in the initial screen (ROC- 325, GSK2578215A, DC661, MRT68921 HCl, Lucanthone, CZC-25146, GNE7915, edaravone, and LY2940020).

We thank the reviewer for these points. We have now looked at morphology (birefringence), swim behavior (via touch evoked escape response), and autophagy (using western blot analyses). The autophagy studies are now included as Figure EV 1.

Additional points:

1. Western Blot in Fig. 5c does not look really convincing. Also, why are the bands at different heights?

We have repeated these data. A new blot is now included. There is some variation in band location that is likely technical in nature. The LCI band is faint but definitely detected in *vma21* mutants. We also now include an orthogonal approach for looking at autophagy – measurement of LC3 from a transgenic zebrafish line.

2. It is not clear when the liver phenotype was studied. This should be looked at during late time points as it might take time point for the liver phenotype to develop.

We thank the reviewer for this suggestion. We have now looked at liver related phenotypes, and uncovered abnormalities consistent with a cholestatic liver disease in the *vma21* deficient zebrafish. This is similar to what has been observed in patients.

3. *Vma21* patients with liver disease also have glycosylation defects. If possible in fish, this should also be studied.

We were not able to figure out how to do these glycosylation studies, though we appreciate that it would be interesting to evaluate.

Referee #3 (Comments on Novelty/Model System for Author):

Pathological characterization of the zebrafish model is not enough while the disease is an entity mainly defined by pathological features in human patients (see the "Remarks to be sent to the author").

Ultimately XMEA is a genetic disease caused by *VMA21* mutations that reduce/abrogate gene expression and/or function. Our model well recapitulates the underlying genetics in terms of *vma21* mutation, and thus is a faithful model from that prospective. The presence of aberrant autophagy is a pathologic hallmark of MEAs as a group of myopathies, and we do observe autophagic vacuoles on EM and evidence of disrupted autophagy by biochemical analyses and through an LC3 transgenic fish. However, we did not observed the LAMA2/DMD/compliment positive vacuoles as reported in patient biopsies. We used immunostaining with anti-laminin and anti-dystrophin to try and identify these. Whether the lack of these is reflective of species differences, or instead reflective of the age(s) at which we were able to analyze muscle in our model, is not clear. We have added a sentence to the discussion about this limitation.

Referee #3 (Remarks for Author):

Huang et al report a novel zebrafish model of X-linked myopathy with excessive autophagy (XMEA), which is actually the first animal model of the disease, and the authors claim that their model replicates most features of the human disease. Furthermore, the authors identified that edaravone and LY294002 improve the phenotype.

This is an important paper but I have two major concerns.

1) I am afraid that the pathological characterization of the phenotype is insufficient considering that XMEA is an entity mainly defined by pathological features. In human, XMEA is pathologically characterized by: 1) autophagic vacuoles with unique sarcolemmal features (AVSF), 2) multilayered basal lamina with exocytosed materials in between the layers, 3) the deposition of calcium and C5b-9 on the sarcolemma; while the authors have shown only the presence of autophagic vacuoles. Pathological features of the model zebrafish should be characterized in more details focusing on the abovementioned features.

See above regarding the fact that we did not detect complement/DMD/merosin positive vacuoles (though we did see double membrane autophagic vacuoles, the defining

pathologic feature of autophagic vacuoles). The reason for this is not clear and may either be a species difference or else be due to the age of muscle at the time of sampling. Of note, a recent natural history study observed AVSF/complement in only 60% of biopsies (Fernandez-Eulate et al., J Neurology, 2024), suggesting they are not an invariant feature of the disease.

2) I wonder what would be the mechanism of edaravone and LY294002 to improve the phenotype. The authors use the term "autophagy antagonist" but I wonder if the phenotype improvement is truly due to the inhibition of autophagy. This should be assessed in more details for example by knocking down essential genes in autophagy in the zebrafish model. In the same line, the authors may want to evaluate acidification status in the treated zebrafish.

We appreciate these points, and have looked into them further. LY294002 reduced the amount of p62 and the amount of LC1/II, consistent with its known role in inhibiting Pik3c3 and preventing the initiation and early phases of autophagy. With edaravone, we did not see a change with LC3, but did see a decrease of p62, so the exact impact on autophagy (at least at the whole animal level) was less clear. These data are now added as Figure EV 1.

1st Oct 2024

Dear Dr. Dowling,

Thank you for the submission of your revised manuscript to EMBO Molecular Medicine. I am pleased to inform you that we will be able to accept your manuscript pending the following final amendments:

- 1) Please address two points raised by the referee #2.
- 2) Figures: Main figures should be removed from the manuscript and uploaded as individual, high resolution figure files. Legends should be compiled at the end of the manuscript text. Fig EV1 should be uploaded as a high resolution figure file, and its legend should be added to the manuscript, after the main figure legends and under the heading "Expanded View Figure Legends".
- 3) In the main manuscript file, please do the following:
 - Please address all comments suggested by our data editors listed below:
 - o Figure legends:
 1. Please note that the data availability statement is not provided in the manuscript.
 2. Please note that the exact p values are not provided in the legends of figures 1d; 2e-f, h-m; 3c; 4d, i; 5c; EV 1b, d, f.
 3. Please note that information related to n is missing in the legends of figures EV 1b, d, f.
 4. Please note that the error bars are not defined in the legends of figures EV 1b, d, f.
 - Reduce keywords to max. 5.
 - Remove data not shown (p.40).
 - Add callouts for Fig 1B, Fig 6C and D. There is a callout for a Fig 7, which does not exist.
 - Add "Disclosure and competing interests statement". We updated our journal's competing interests policy in January 2022 and request authors to consider both actual and perceived competing interests. Please review the policy <https://www.embopress.org/competing-interests> and update your competing interests if necessary.
 - Author contributions: Please remove it from the manuscript and specify author contributions in our submission system. CRediT has replaced the traditional author contributions section because it offers a systematic machine-readable author contributions format that allows for more effective research assessment. Please use the free text boxes beneath each contributing author's name to add specific details on the author's contribution. More information is available in our guide to authors: <https://www.embopress.org/page/journal/17574684/authorguide#authorshipguidelines>
 - Indicate in legends number and nature of replicates and exact p= values, not a range, along with the statistical test used. To keep the figures "clear" some authors found providing an Appendix table Sx with all exact p-values preferable. You are welcome to do this if you want to.
- 4) Tables: Please combine appendix tables S2, S4, S5, S7, S9, S10, S11, S12, S13, S14 and S15 in an Appendix and upload it as a single PDF file. Tables should be renumbered to Appendix Table S1 etc. and add table of content with page numbers on the title page of the Appendix file. Tables S1, S3, S6, S8 should be renamed to Table EV1 etc. Please also update table callouts in the main manuscript text.
- 5) The Paper Explained: Please provide "The Paper Explained" and add it to the main manuscript text. Please check "Author Guidelines" for more information. <https://www.embopress.org/page/journal/17574684/authorguide#researcharticleguide>
- 6) Synopsis: Every published paper now includes a 'Synopsis' to further enhance discoverability. Synopses are displayed on the journal webpage and are freely accessible to all readers. They include separate synopsis image and synopsis text.
 - Synopsis image: Please provide a visual abstract as a high-resolution jpeg file 550 px-wide x (300-600)-px high to illustrate your article.
 - Synopsis text: Please provide a short standfirst (maximum of 300 characters, including space) as well as 2-5 one sentence bullet points that summarise the paper as a .doc file. Please write the bullet points to summarise the key NEW findings. They should be designed to be complementary to the abstract - i.e. not repeat the same text. We encourage inclusion of key acronyms and quantitative information (maximum of 30 words / bullet point). Please use the passive voice.
 - Please check your synopsis text and image before submission with your revised manuscript. Please be aware that in the proof stage minor corrections only are allowed (e.g., typos).
- 7) As part of the EMBO Publications transparent editorial process initiative (see our Editorial at <http://embomolmed.embopress.org/content/2/9/329>), EMBO Molecular Medicine will publish online a Review Process File (RPF) to accompany accepted manuscripts. This file will be published in conjunction with your paper and will include the anonymous referee reports, your point-by-point response and all pertinent correspondence relating to the manuscript. Let us know whether you agree with the publication of the RPF and as here, if you want to remove or not any figures from it prior to publication. Please note that the Authors checklist will be published at the end of the RPF.
- 8) Please provide a point-by-point letter INCLUDING my comments as well as the reviewer's reports and your detailed responses (as Word file).

I look forward to reading a new revised version of your manuscript as soon as possible.

Yours sincerely,

Zeljko Durdevic

*** Instructions to submit your revised manuscript ***

- 1) a .docx formatted version of the manuscript text (including Figure legends and tables)
- 2) Separate figure files*
- 3) supplemental information as Expanded View and/or Appendix. Please carefully check the authors guidelines for formatting Expanded view and Appendix figures and tables at <https://www.embopress.org/page/journal/17574684/authorguide#expandedview>
- 4) a letter INCLUDING the reviewer's reports and your detailed responses to their comments (as Word file).
- 5) The paper explained: EMBO Molecular Medicine articles are accompanied by a summary of the articles to emphasize the major findings in the paper and their medical implications for the non-specialist reader. Please provide a draft summary of your article highlighting
 - the medical issue you are addressing,
 - the results obtained and
 - their clinical impact.This may be edited to ensure that readers understand the significance and context of the research. Please refer to any of our published articles for an example.
- 6) Author contributions: the contribution of every author must be detailed in a separate section.
- 7) EMBO Molecular Medicine now requires a complete author checklist (<https://www.embopress.org/page/journal/17574684/authorguide>) to be submitted with all revised manuscripts. Please use the checklist as guideline for the sort of information we need WITHIN the manuscript. The checklist should only be filled with page numbers where the information can be found. This is particularly important for animal reporting, antibody dilutions (missing) and exact values and n that should be indicated instead of a range.
- 8) Every published paper now includes a 'Synopsis' to further enhance discoverability. Synopses are displayed on the journal webpage and are freely accessible to all readers. They include a short stand first (maximum of 300 characters, including space) as well as 2-5 one sentence bullet points that summarise the paper. Please write the bullet points to summarise the key NEW findings. They should be designed to be complementary to the abstract - i.e. not repeat the same text. We encourage inclusion of key acronyms and quantitative information (maximum of 30 words / bullet point). Please use the passive voice. Please attach these in a separate file or send them by email, we will incorporate them accordingly.

You are also welcome to suggest a striking image or visual abstract to illustrate your article. If you do please provide a jpeg file 550 px-wide x 300-600px high.

9) A Conflict of Interest statement should be provided in the main text

10) Please note that we now mandate that all corresponding authors list an ORCID digital identifier. This takes <90 seconds to complete. We encourage all authors to supply an ORCID identifier, which will be linked to their name for unambiguous name identification.

Currently, our records indicate that the ORCID for your account is 0000-0002-3984-4169.

Link Not Available

11) Include a Reagents and Tools Table as part of the Methods section, which can be downloaded from our author guidelines (<https://www.embopress.org/page/journal/17574684/authorguide#structuredmethods>)

Photos 400-800 DPI

*Additional important information regarding figures and illustrations can be found at

<https://bit.ly/EMBOPressFigurePreparationGuideline>. See also figure legend preparation guidelines:

<https://www.embopress.org/page/journal/17574684/authorguide#figureformat>

***** Reviewer's comments *****

Referee #1 (Remarks for Author):

Is suitable for publication.

Referee #2 (Remarks for Author):

Thanks for addressing my comments. I still have two points:

1. The LC3B blot in Fig. 4c is still not convincing, as the LC3I is so faint. It is also strange that the bands are at different heights. Is it possible to include a positive control that induces both LC3I and II? In cell culture, Bafilomycin is typically used for this.
2. Thanks for looking more deeply into a potential liver phenotype. Would it be possible to perform some histologic analysis of the liver? This is important because in some of the Vma21 patients a liver steatosis was observed. Please also cite the liver phenotypes of patients with mutations in other V-ATPase assembly factors (PMID: 27231034, PMID: 29127204).

We thank the editor and the reviewers for the evaluation of our revised manuscript, and for their recommendation of acceptance pending minor response to reviewers. We have been able to address all of the updated comments from reviewer 2, and have now included an updated manuscript that has been strengthened by the inclusion of new data and discussion. Point by point responses to reviewer critiques are included below.

Response to Reviewer 2:

1. The LC3B blot in Fig. 4c is still not convincing, as the LC3I is so faint. It is also strange that the bands are at different heights. Is it possible to include a positive control that induces both LC3-I and II. In cell culture Bafilomycin is typically used for this.

Response: We agree with the reviewer and have re-run those particular LC3B western blots with fresh protein lysates. Admittedly, using whole fish larvae lysates would display more background western blot images than isolated cell line immunoblots for LC3B. Nevertheless, we have obtained clearer blot images that reflect the impaired autophagy that is occurring in the *vma21* mutant zebrafish.

For a positive control, we utilized a dose-escalation ethanol (EtOH)-treatment of WT (*AB* strain) zebrafish treatment (Wen *et al.*, *Life*, 2022; PMID: 9410481) that has been shown to induce whole body autophagy (Varga *et al.*, *Methods*, 2015; PMID: 25498006). We observed increases in total autophagy with increases in ethanol toxicity past the 0.5% ethanol limit as previously reported (Wen *et al.*, *Life*, 2022; PMID: 9410481). We present this proof-of-concept positive control here, and have also included a description in the methods and the data as Expanded View Figure 1 (**Figure EV1**).

2. Thanks for looking more deeply into the potential liver phenotype. Would it be possible to perform some histological analysis of the liver? This is more important because in some of the *Vma21* patients a liver steatosis was observed. Please also cite the liver phenotypes of patients with mutations in other V-ATPase assembly factors (PMID: 27231034, PMID: 29127204).

Response: As per the reviewer's request, we have added H&E staining, which shows fat droplets in the *vma21* mutant zebrafish suggestive of hepatic steatosis, similar to that seen in some patients. We have also added quantification of liver size wherein we show that mutants have smaller liver size as compared to controls, further highlighting the liver injury phenotype with mutations in *vma21*. Of note, to perform liver size analysis, we generated *vma21* "Crispant" mutants in the background of an *fabp:mCherry* transgenic line (which produced a red fluorescent liver). We have also cited the papers which mention liver phenotypes of patients with mutations in other V-ATPase assembly factors.

Editorial comments

We have modified the manuscript and figures to comply with the journal specifications. We have also now included the "paper explained" section and the acknowledgement section. We thank the editor and editorial team for all of the assistance and guidance with our manuscript.

10th Feb 2025

Dear Dr. Dowling,

We are pleased to inform you that your manuscript is accepted for publication and is now being sent to our publisher to be included in the next available issue of EMBO Molecular Medicine.

Zeljko Durdevic
Senior Editor
EMBO Molecular Medicine
